# The $N^6$-methyladenosine methyltransferase METTL16 enables erythropoiesis through safeguarding genome integrity

Masanori Yoshinaga [1] ✉, Kyuho Han [2], David W. Morgens[2], Takuro Horii[3], Ryosuke Kobayashi[3], Tatsuaki Tsuruyama[4], Fabian Hia [1], Shota Yasukura[1], Asako Kajiya[1], Ting Cai[1], Pedro H. C. Cruz[5], Alexis Vandenbon [6], Yutaka Suzuki[7], Yukio Kawahara [5], Izuho Hatada[3,8], Michael C. Bassik [2] & Osamu Takeuchi [1] ✉

During erythroid differentiation, the maintenance of genome integrity is key for the success of multiple rounds of cell division. However, molecular mechanisms coordinating the expression of DNA repair machinery in erythroid progenitors are poorly understood. Here, we discover that an RNA $N^6$-methyladenosine (m6A) methyltransferase, METTL16, plays an essential role in proper erythropoiesis by safeguarding genome integrity via the control of DNA-repair-related genes. METTL16-deficient erythroblasts exhibit defective differentiation capacity, DNA damage and activation of the apoptotic program. Mechanistically, METTL16 controls m6A deposition at the structured motifs in DNA-repair-related transcripts including *Brca2* and *Fancm* mRNAs, thereby upregulating their expression. Furthermore, a pairwise CRISPRi screen revealed that the MTR4-nuclear RNA exosome complex is involved in the regulation of METTL16 substrate mRNAs in erythroblasts. Collectively, our study uncovers that METTL16 and the MTR4-nuclear RNA exosome act as essential regulatory machinery to maintain genome integrity and erythropoiesis.

Erythropoiesis is the multistep process that produces red blood cells from hematopoietic stem cells (HSCs)[1–3]. This is a sophisticated process orchestrated by various mechanisms, including transcriptional, epigenetic, post-transcriptional, and post-translational regulations[4,5]. In humans, erythropoiesis generates 200 billion new erythrocytes every day, which are essentially required for oxygen delivery[6]. To meet this demand, erythroid differentiation is coupled with rapid expansion to produce a large number of erythrocytes. Erythroid cells undergo multiple rounds of DNA replication and cell division during their differentiation[7]. The failure or insufficiency of erythropoiesis leads to anemia, a common health problem that affects over half of a billion people worldwide[8]. Moreover, dysfunction of erythropoiesis is seen in many hematological diseases. Therefore, a better understanding of the molecular basis of erythropoiesis is necessary for the treatment of anemia and other hematological disorders.

During erythropoiesis, erythroid cells incorporate a large amount of transferrin-bound iron from the bloodstream, thereby producing

[1]Department of Medical Chemistry, Graduate School of Medicine, Kyoto University, Kyoto 606-8501, Japan. [2]Department of Genetics, Stanford University School of Medicine, Stanford, CA 94305, USA. [3]Laboratory of Genome Science, Biosignal Genome Resource Center, Institute for Molecular and Cellular Regulation, Gunma University, Gunma 371-8512, Japan. [4]Department of Drug and Discovery Medicine, Graduate School of Medicine, Kyoto University, Kyoto 606-8501, Japan. [5]Department of RNA Biology and Neuroscience, Graduate School of Medicine, Osaka University, Osaka 565-0871, Japan. [6]Laboratory of Tissue Homeostasis, Institute for Life and Medical Sciences, Kyoto University, Kyoto 606-8507, Japan. [7]Laboratory of Functional Genomics, Department of Medical Genome Sciences, Graduate School of Frontier Sciences, The University of Tokyo, Chiba 277-8562, Japan. [8]Viral Vector Core, Gunma University Initiative for Advanced Research (GIAR), Gunma 371-8512, Japan. ✉e-mail: m_yoshi@mfour.med.kyoto-u.ac.jp; otake@mfour.med.kyoto-u.ac.jp

the oxygen carrier hemoglobin. Erythroid cells express high levels of transferrin receptor (TfR1 or CD71), thereby enabling the incorporation of transferrin-bound iron[6,9,10]. TfR1 expression is greatly upregulated when the cells enter the late erythroid progenitors (CFU-E) stage, and expression gradually decreases during maturation by proteolytic shedding[11]. This upregulation is primarily mediated by the activation of erythroid transcription factors, including GATA binding protein 1 (GATA-1)[9], while post-transcriptional mechanisms also operate to regulate TfR1 expression[6,12,13]. TfR1 regulators, including GATA-1 and Iron regulatory protein 2 (IRP2), play critical roles in erythroid differentiation and function[14–16]. Therefore, TfR1 is a signature molecule coupled with erythroid differentiation and is heavily controlled by multiple regulators.

Erythroid cells rapidly undergo multiple cell divisions and simultaneously incorporate iron during differentiation, endangering them to DNA damage[17–20]. Insults to genome integrity lead to cell cycle arrest or apoptosis if irreparable, causing defects in erythropoiesis. Moreover, damaged DNA is sensed by cyclic GMP-AMP synthase (cGAS), which triggers the activation of an adaptor molecule Stimulator of interferon genes (STING, also known as TMEM173)[21–23]. The STING signaling leads to the production of type I interferons (IFNs), which inhibits erythroid differentiation[24–26]. Since deletion of DNA-repair machineries in vivo results in erythropoietic failure[27,28], the rigorous maintenance of genome integrity is an important prerequisite for proper erythroid differentiation, survival, and proliferation. However, it is poorly understood how erythroid cells cope with the increased burden of genomic insults.

In this study, by employing a CRISPR-based screen for TfR1 regulators, and by investigating the in vivo roles of the RNA methyltransferase METTL16, we reveal that METTL16-mediated mRNA methylation plays a critical role in the maintenance of genome integrity, thereby enabling erythroid differentiation.

## Results

### A genome-wide CRISPR screen identifies RNA-binding proteins as TfR1 regulators

We hypothesized that genes essential for erythroid differentiation are enriched in the regulators of TfR1 expression. Therefore, we performed a functional CRISPR screen for identifying regulators of TfR1 expression at a genome-wide scale[29] (Methods). The screen was performed using Cas9-expressing K562 cells (K562-Cas9), a human erythroleukemia cell line[29,30] expressing relatively high levels of *TfR1* mRNA (Supplementary Fig. 1a). We transduced K562-Cas9 cells with a whole-genome lentiviral guide RNA (gRNA) library[31], and sorted the cells which expressed high or low levels of TfR1 (CD71) (Fig. 1a, b). We analyzed the enrichment of gRNAs in each sorted fraction using casTLE pipeline[32]. Then we successfully recovered a set of genes that are known to be important for intracellular iron regulation, including *IREB2* (encoding IRP2), *SLC40A1* (Ferroportin or FPN1), and *SLC11A2* (DMT1) (Supplementary Fig. 1b, c and Supplementary Data 1).

The CRISPR screen-detected 894 gene hits as TfR1 regulators using a 1% false discovery rate (FDR). In these hits, gene ontologies (GOs) related to RNA binding and modification were enriched (Fig. 1c and Supplementary Fig. 1d), suggesting that post-transcriptional regulation plays a critical role in TfR1 expression. Therefore, we explored the TfR1-regulatory role(s) of genes annotated to be RNA-binding proteins (RBPs)[33,34]. We picked up individual gRNAs targeting several novel RBPs which influenced TfR1 expression based on their enrichment scores (Fig. 1d and Supplementary Fig. 1e). We chose two of the best-working gRNAs and performed CRISPR-based knockouts for each gene. The knockout results for each candidate gene confirmed that the ablation of RBPs such as METTL16 and FASN led to a great reduction of surface TfR1 expression, to a similar extent observed in the ablation of IRP2, which is a well-characterized TfR1

regulator (Fig. 1e, f). Conversely, the deficiency of RBPs such as HNRNPL and XPO5 greatly increased TfR1 expression (Fig. 1g). These findings were corroborated by immunoblotting of TfR1 (Supplementary Fig. 1f). Taken together, our CRISPR screen revealed multiple RBPs that regulate TfR1 expression, which has not been implicated in the regulation of erythroid cells.

### METTL16 regulates TfR1 expression and erythroid differentiation in a methyltransferase activity-dependent manner

We focused on an RNA $N^6$-methyladenosine (m6A) methyltransferase enzyme METTL16 for further functional analyses since METTL16 ablation greatly reduced TfR1 expression (Fig. 1f, Supplementary Fig. 1e). The m6A methylation, the most prevalent and abundant RNA internal modification, regulates many aspects of RNA fate and influences many biological processes such as development and cell differentiation[35–40]. METTL16 has been identified as an evolutionarily-conserved m6A writer for U6 snRNA[41–45] and mRNAs of S-adenosyl methionine (SAM) synthases, which regulate the intracellular SAM concentration[41,42,44]. In addition, METTL16 is reported to interact with non-coding RNAs (ncRNAs), ribosomal RNAs, and nascent mRNAs[45–47].

The screen results were further validated by the depletion of METTL16 expression via the CRISPR interference (CRISPRi) system, leading to a drastic decrease in TfR1 expression under the iron-depleted condition, which minimizes the change in intracellular iron levels (Supplementary Fig. 2a–c). *TfR1* mRNA levels were also downregulated in METTL16-deficient cells (Supplementary Fig. 2d). Moreover, METTL16 is required for hemin-induced *HBG* mRNA expression, which represents the erythroid differentiation of K562 cells (Supplementary Fig. 2e). The regulation of TfR1 expression and erythroid differentiation is mediated by METTL16's methyltransferase activity, since the reconstitution of wild-type METTL16, but not the catalytic-dead mutants (PP185/186AA, F187G)[41], restored surface TfR1 expression and hemin-induced *HBG* mRNA expression in METTL16-deficient cells (Fig. 1h, i, Supplementary Fig. 2f). Altogether, our CRISPR screen identified the RNA methyltransferase METTL16 as a critical regulator of TfR1 expression and erythroid differentiation.

### METTL16 regulates erythroid differentiation in vivo

We next turned to the role of METTL16 in hematopoiesis in vivo. Among hematopoietic lineages, megakaryocyte-erythroid progenitors (MEP) and Ter119+ erythroid cells, both of which are progenitors of erythrocytes, express high levels of *Mettl16* mRNA compared to other hematopoietic cell types (Fig. 2a, Supplementary Fig. 3). Analysis of a single-cell RNA-seq database of erythroid lineages[48] confirmed the upregulation of *Mettl16* mRNA in the midst of erythroid differentiation in the mouse fetal liver (FL) and bone marrow (Fig. 2b).

To investigate the in vivo role of METTL16 in erythropoiesis, we generated mice that lack METTL16, specifically in erythroblasts using erythropoietin receptor (Epor) promoter-driven Cre transgenic mice (*Mettl16*fl/fl*Epor-Cre*+ mice, Supplementary Fig. 4a, b). *Mettl16*fl/fl*Epor-Cre*+ mice showed prenatal lethality, and all the fetuses did not survive past embryonic day 13.5 (E13.5), when definitive erythropoiesis takes over primitive erythropoiesis (Fig. 2c). Consistent with this finding, *Mettl16*fl/fl*Epor-Cre*+ embryos appeared pale on E11.5–12.5, suggesting a defect in erythropoiesis (Fig. 2d).

Since *Epor* promoter-driven Cre activity is observed in primitive and definitive erythroid cells, we next investigated primitive erythroid cells in the E10.5 peripheral blood. We found that surface TfR1 expression was downregulated under METTL16 deficiency (Fig. 2e–g), consistent with our initial CRISPR screen result. Nevertheless, no macroscopic change in fetuses was observed on E10.5. Therefore, we next investigated definitive erythropoiesis during the later stage of prenatal development. Flow cytometric analyses of E11.5 FL cells revealed that erythroblast differentiation was greatly impaired in

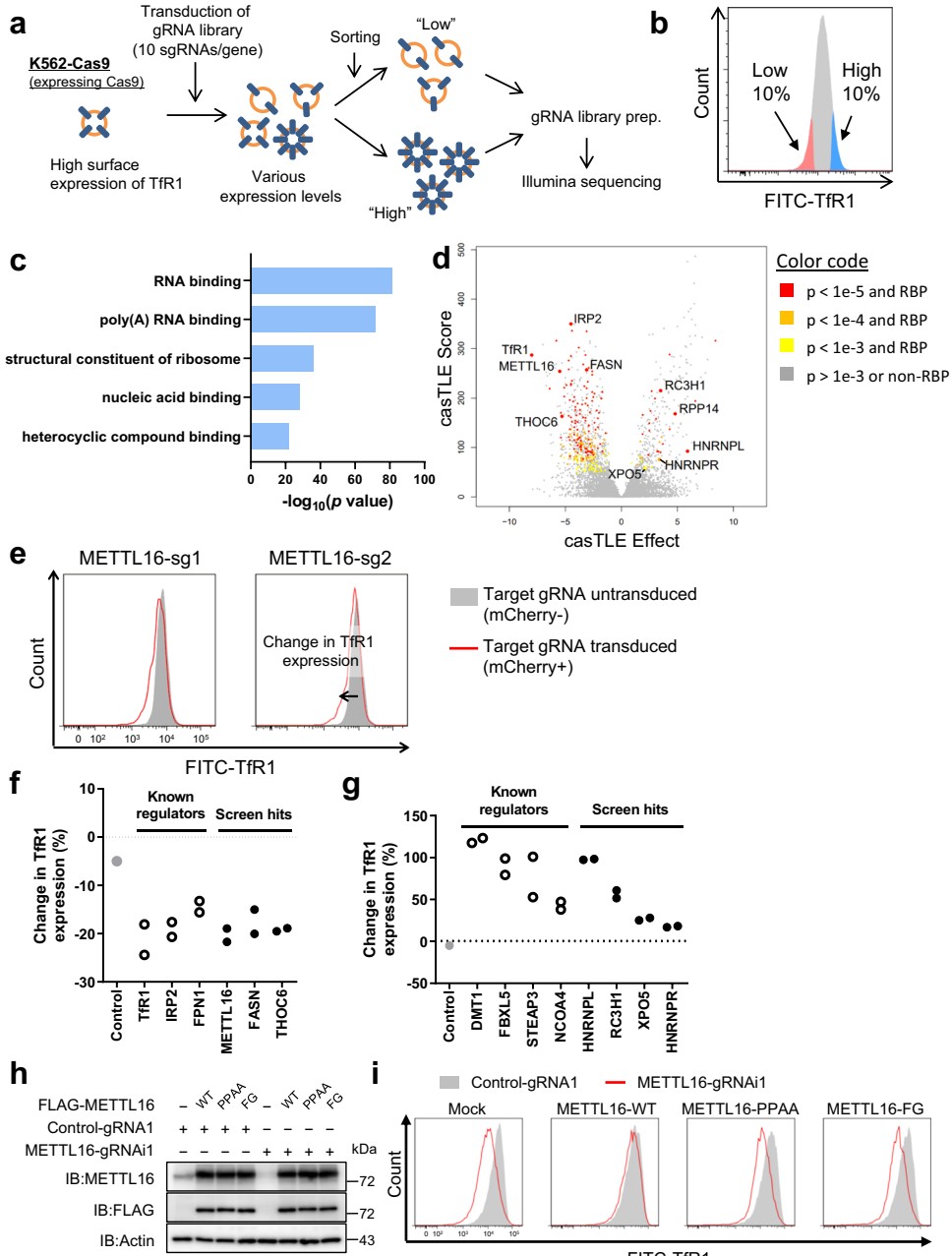

**Fig. 1 | A genome-wide pooled CRISPR screen identifies novel RNA-binding proteins which regulate TfR1 expression. a**, **b** Workflow (**a**) and cell sorting strategy (**b**) of the genome-wide CRISPR screen for TfR1 regulators. **c** Gene ontology analysis of the TfR1 CRISPR screen hits. Statistical analysis was performed using Immuno-Navigator[83]. **d** Volcano plot of TfR1 phenotype (casTLE Effect, the most likely effect size as determined by casTLE) and the confidence in that effect size (casTLE Score) from the CRISPR screen. The screen was performed in duplicate. **e**–**g** Validation of the screen hits. Two gRNAs per gene were chosen for retests. K562-Cas9 cells were transduced with lentiviral vectors encoding individual gRNAs together with mCherry. Then cells were stained with anti-TfR1 antibody and analyzed using flow cytometry (**e**). Change in TfR1 expression levels between untransduced (mCherry−, gray) and transduced (mCherry+, red) were calculated and summarized (**f**, **g**). Each symbol represents the results of individual gRNAs. Data were pooled from two independent experiments. **h**, **i** Effects of wild-type or catalytic-dead mutant METTL16 (PP185/186AA, F187G) on surface TfR1 expression in the presence of DFO. Similar expression levels of wild-type or mutant METTL16 protein were confirmed (**h**). Surface TfR1 expression levels of cells expressing indicated gRNAs and proteins were examined by flow cytometry (**i**). Similar results were obtained in two independent experiments. Source data are provided as a Source Data file.

*Mettl16*[fl/fl]*Epor-Cre*[+] mice, especially at the transition from CD71[+]Ter119[−] (proerythroblasts and basophilic erythroblasts) to CD71[+]Ter119[+] (mostly polychromatic erythroblasts) stage (Fig. 2h). We confirmed that the Cre-mediated recombination of *Mettl16*-floxed allele began in CD71[+]Ter119[−] erythroblasts and was almost completed in CD71[+]Ter119[+] erythroblasts (Supplementary Fig. 4c). Cytospin preparation of the FL cells revealed the increased proerythroblasts and basophilic erythroblasts, and reduced but enlarged polychromatic erythroblasts under

METTL16 deficiency (Fig. 2i). Consistently, forward scatter (FSC) values, which is proportional to cell size, were higher in CD71[+]Ter119[+] erythroblasts from *Mettl16*[fl/fl]*Epor-Cre*[+] mice compared with controls (Fig. 2j). Since erythroid cells are known to gradually reduce their cell size during differentiation, these findings suggest that METTL16 deficiency impairs and/or delays erythroid differentiation. Moreover, erythroblast differentiation was greatly impaired under METTL16 deficiency on E12.5, although there is some progression in

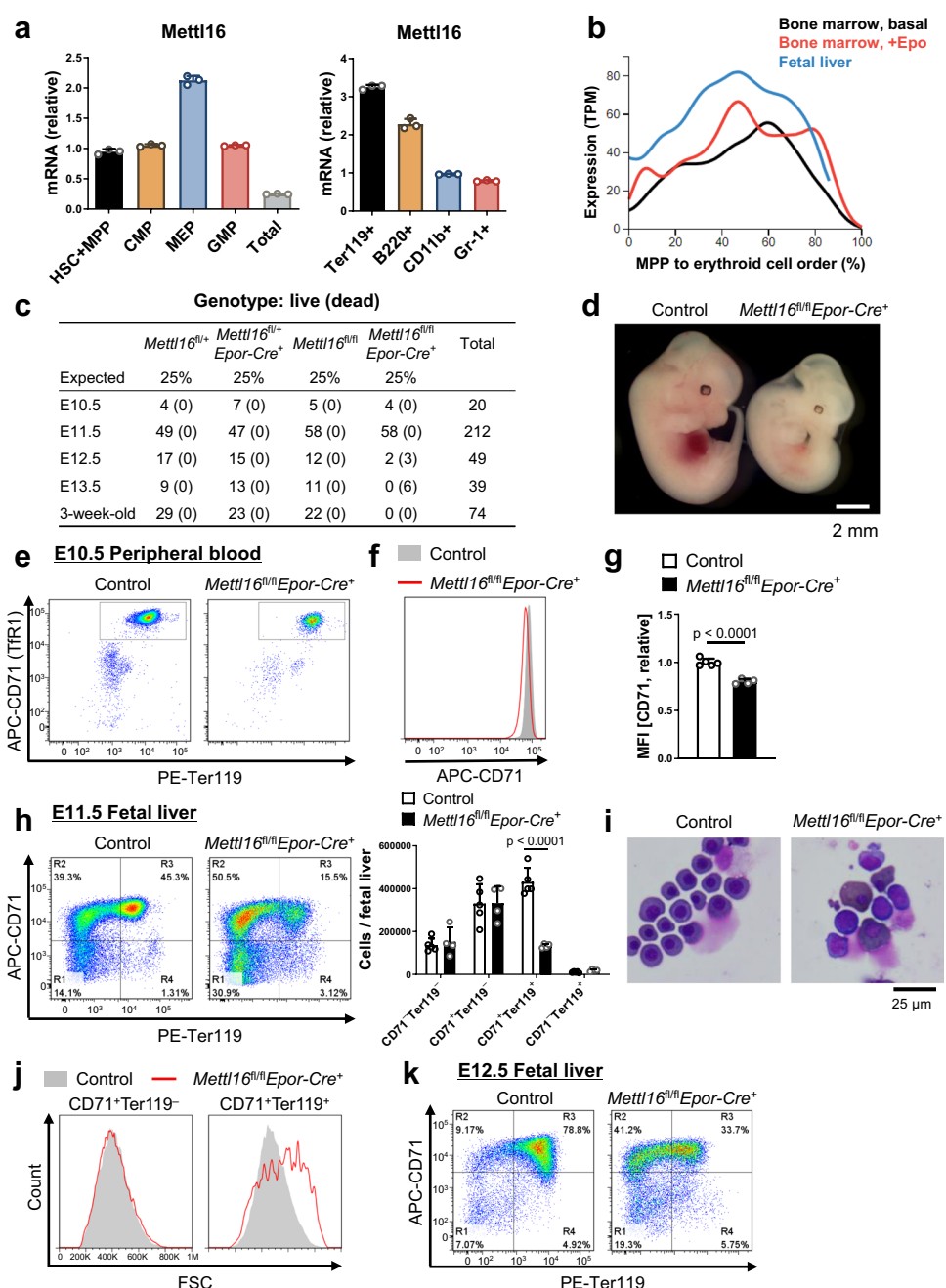

**Fig. 2 | RNA methyltransferase METTL16 regulates erythroid differentiation in vivo. a** *Mettl16* mRNA expression levels in each hematopoietic compartment (*n* = 3 technical replicates). HSC hematopoietic stem cells, MPP multipotent progenitors, CMP common myeloid progenitors, GMP granulocyte-monocyte progenitors, MEP megakaryocyte-erythrocyte progenitors. **b** Gene expression patterns of *Mettl16* mRNA during erythroid differentiation generated from the single-cell RNA-seq analysis[48]. **c** Survival rates of mice with each genotype. **d** Fetuses on E12.5 of control and *Mettl16*^fl/fl^*Epor-Cre*^+^ mice. **e–g** Flow cytometric analysis of the E10.5 peripheral blood from *Mettl16*^fl/fl^*Epor-Cre*^+^ mice (**e**). The histogram of CD71 intensity

(**f**) and the relative CD71 mean fluorescence intensity (MFI) in the gated population (**g**, *n* = 4–5 mice). Data were pooled from two independent experiments. **h** Flow cytometric analysis of the E11.5 FL from *Mettl16*^fl/fl^*Epor-Cre*^+^ mice (*n* = 4–5 mice). **i** May-Grünwald-Giemsa stain of cytospin slides of E11.5 FL cells from *Mettl16*^fl/fl^*Epor-Cre*^+^ mice. **j** The histogram of FSC intensity in each population in **h**. **k** Flow cytometric analysis of the E12.5 FL from *Mettl16*^fl/fl^*Epor-Cre*^+^ mice. Data are expressed as mean ± SD (**a**, **g**, **h**). The *p*-values were calculated using two-tailed Student's *t*-test (**g**, **h**). Images are representative of samples obtained from at least three mice (**d** and **i–k**). Source data are provided as a Source Data file.

differentiation compared to E11.5 (Fig. 2k). Taken together, these results indicate that METTL16 is critical for erythroid differentiation in vivo.

## METTL16 is required for proper erythroid gene expression in vivo

To gain insights into the mechanism(s) regulating erythroid differentiation by METTL16, we performed the transcriptome analysis. In

CD71^+^Ter119^−^ stage, the change in transcriptome under *Mettl16* deficiency was modest, and most of the differentially regulated genes were associated with the hypoxic response, which was attributable to severe anemia in *Mettl16*^fl/fl^*Epor-Cre*^+^ mice (Supplementary Fig. 4d and Supplementary Data 2). In contrast, we observed a drastic alteration in the transcriptome in *Mettl16*-deficient CD71^+^Ter119^+^ erythroblasts (Supplementary Fig. 4d and Supplementary Data 3). The expression of erythroid-related genes, including *TfR1* and *Klf1*

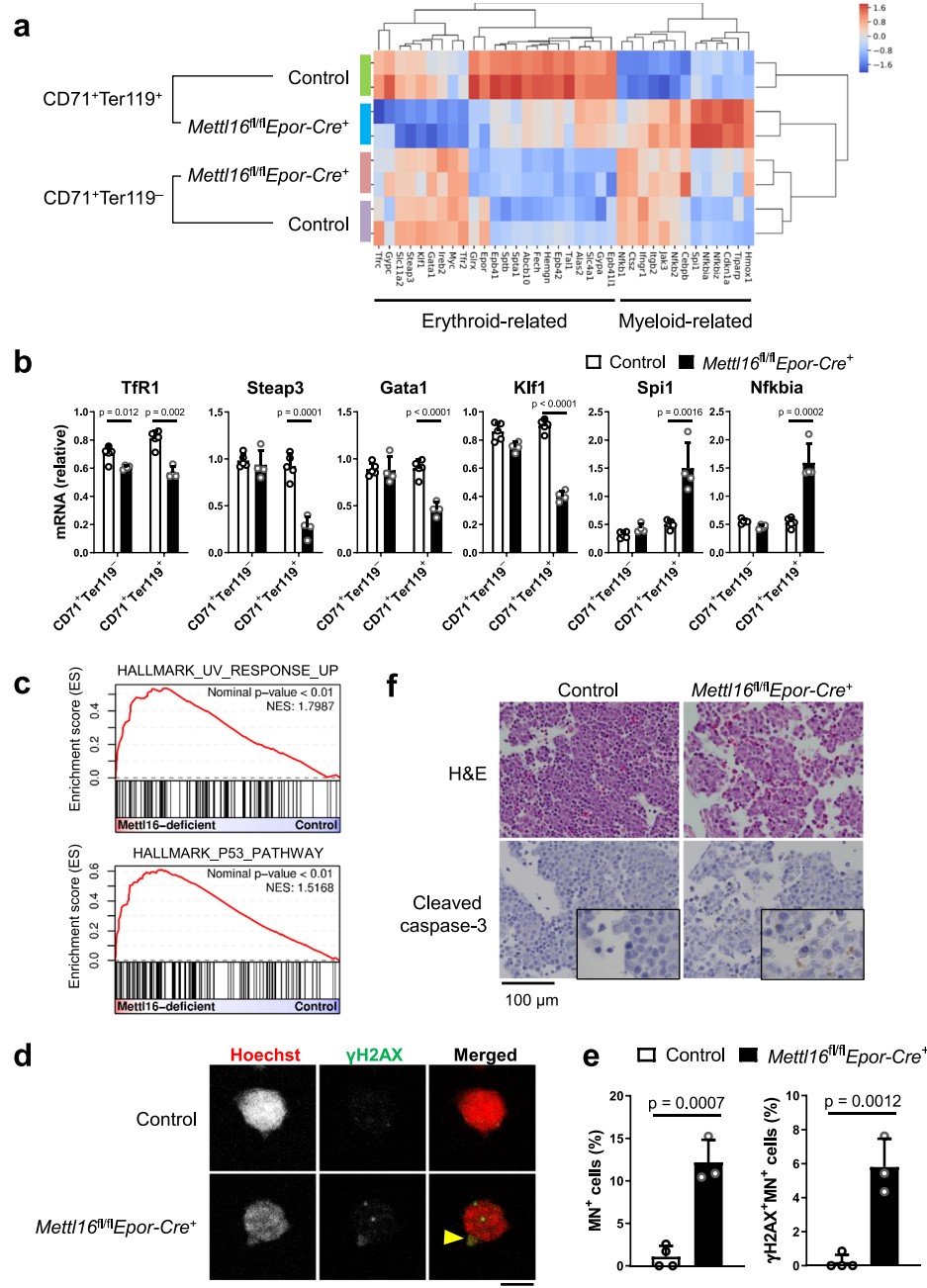

**Fig. 3 | METTL16 is required for maintenance of erythroid identity and genome integrity. a** Heatmap showing the relative mRNA expression levels of erythroid-related and myeloid-related genes in the erythroblasts from E11.5 control and *Mettl16*fl/fl*Epor-Cre*+ mice. **b** mRNA expression levels of erythroid-related and myeloid-related genes in the erythroblasts from control (*n* = 5 mice) and E11.5 *Mettl16*fl/fl*Epor-Cre*+ mice (*n* = 4 mice). mRNA expression levels of genes were normalized to the expression level of β-actin (*Actb*). **c** GSEA plot showing the enrichment of the hallmarks of UV response and p53 pathway in erythroblasts from E11.5 *Mettl16*fl/fl*Epor-Cre*+ mice. Statistical analysis was performed using GSEA software. **d, e** CD71+Ter119+ erythroblasts of the FL from E11.5 control and *Mettl16*fl/fl*Epor-Cre*+ mice stained by Hoechst 33342 (red) and anti-γH2AX antibody (green) and

analyzed by the confocal microscopy. Representative images of erythroblasts harboring γH2AX-positive micronuclei (arrowhead, **d**). The enumeration of micronuclei (MN) and γH2AX-positive MN containing cells. The positive cell numbers were divided by all of the analyzed cell numbers (**e**, *n* = 3–4 mice pooled from three independent experiments). **f** Hematoxylin and eosin (H&E, top) and immunohistochemical staining of cleaved caspase-3 (bottom) of the FL from E11.5 control and *Mettl16*fl/fl*Epor-Cre*+ mice. Data are expressed as mean ± SD (**b**, **e**). The *p*-values were calculated using two-tailed Student's *t*-test (**b**, **e**). Images are representative of samples obtained from three independent experiments (**d**, **f**). Source data are provided as a Source Data file.

were downregulated in *Mettl16*-deficient CD71+Ter119+ erythroblasts, while there was an upregulation of myeloid-related gene expression such as *Spi1* (PU.1), *Cebpb* and *Nfkbia* (Fig. 3a, b). Consistently, GO analysis revealed that *Mettl16*-deficient CD71+Ter119+ erythroblasts exhibited an upregulation and downregulation of genes associated with the immune system process and cell cycle categories,

respectively (Supplementary Fig. 4e, f). The frequency of Ki-67+ proliferating cells was reduced in *Mettl16*-deficient erythroblasts (Supplementary Fig. 4g), suggesting that METTL16 is required for cell division of erythroid cells.

Furthermore, gene set enrichment analysis (GSEA) revealed that genes associated with ultraviolet (UV) response or p53 pathway, which

are relevant to DNA damage, were enriched in the genes upregulated in *Mettl16*-deficient CD71[+]Ter119[+] erythroblasts (Fig. 3c). Therefore, we hypothesized that increased DNA damage in *Mettl16*-deficient erythroblasts is involved in differentiation failure since unrepairable DNA damage leads to cell cycle arrest and/or apoptosis[49]. Cytological analysis revealed that *Mettl16*-deficient erythroblasts contained micronuclei, which are produced upon DNA damage, at a higher frequency (Figs. 2i and 3d). Moreover, many of the micronuclei were positive for γH2AX, a marker of DNA damage (Fig. 3d, e). Immunohistochemical analysis revealed apoptotic erythroblasts with cleaved caspase-3 expression in FL from *Mettl16*[fl/fl]*Epor-Cre*[+] mice (Fig. 3f). It has been shown that caspase proteins cleave GATA-1 during erythroid differentiation[50–52]. Consistently, we observed the downregulation of GATA-1 expression in *Mettl16*-deficient erythroblasts (Supplementary Fig. 5a, b). Since GATA-1 is an essential transcription factor required for erythroid gene expressions such as TfR1 and KLF1, erythroid differentiation might be inhibited by caspase-mediated GATA-1 cleavage under METTL16 deficiency. Collectively, METTL16 plays an essential role in safeguarding genome integrity and erythroid gene expression during erythroblast differentiation.

Micronuclei are known to trigger the cGAS-STING signaling pathway[21–23]. Indeed, genes induced by IFNs and a transcription factor STAT1 were expressed highly in *Mettl16*-deficient CD71[+]Ter119[+] erythroblasts (Supplementary Fig. 5c, d). However, the activation of IFN signaling did not have an influence on erythroid differentiation under METTL16 deficiency, since further ablation of type I IFN receptor or STING failed to restore erythroid differentiation (Supplementary Fig. 5e, f). In summary, these findings suggest that METTL16 licenses proper erythroid differentiation by regulating DNA-damage responses, but not by inhibiting the IFN signaling.

## METTL16 regulates m⁶A deposition on mRNAs associated with DNA-damage responses

We next investigated the molecular mechanism of how METTL16 prevents DNA damage during erythropoiesis. Whereas METTL16 is involved in splice site selection in yeasts[43], analysis of the RNA-seq data using the IsoformSwitchAnalyzeR pipeline[53] revealed that alternative splicing events such as alternative 5′ donor and 3′ site acceptor site selections were not significantly altered in *Mettl16*-deficient CD71[+]Ter119[+] erythroblasts (Supplementary Fig. 6). Therefore, the METTL16-mediated regulation of transcriptome in erythroblasts is likely through mechanisms independent of the global change in alternative splicing.

We next asked if METTL16 directly controls m⁶A deposition on mRNAs. Since the number of erythroblasts directly obtained from *Mettl16*[fl/fl]*Epor-Cre*[+] embryos was insufficient to perform m⁶A profiling, we took advantage of the in vitro erythroblast culture treated with erythropoietin (Epo) and (Z)−4-Hydroxytamoxifen (4-OHT) from the E13.5 FL of *Mettl16*[fl/fl]*CreERT2*[+] mice (Fig. 4a). *Mettl16* mRNA expression was abrogated, and *Mat2a* mRNA, an established target of METTL16[41,42], was downregulated under METTL16 deficiency in the cultured erythroblasts (Fig. 4b). Consistent with the in vivo finding, *Mettl16*-deficient cultured erythroblasts showed the impaired enucleation, which is a final event in erythroid differentiation, although most of the cells were able to enter the CD71[+]Ter119[+] stage (Supplementary Fig. 7a–c). Also, the alkaline comet assay, the gold standard method to detect genomic DNA damage and repair at a single-cell resolution, showed that the tail DNA content, tail length, and tail moment were increased in *Mettl16*-deficient cultured erythroblasts, suggesting that the extent of DNA damage was increased (Supplementary Fig. 8a, b). Moreover, erythroid-related genes were downregulated in *Mettl16*-deficient cultured erythroblasts (Supplementary Fig. 9a). These findings indicate that the *Mettl16*-deficient in vitro cultured erythroid cells recapitulate the defects in maintaining erythroid differentiation and genome integrity observed in vivo.

We then performed methylated RNA immunoprecipitation sequencing (MeRIP-seq) experiments using poly(A)[+] RNA isolated from these cells[54]. MeRIP-seq analysis showed that 3261 sites were differentially methylated, and the majority of them (2839 sites) were hypomethylated under METTL16 deficiency (Fig. 4c and Supplementary Data 4 and 5), consistent with the function of METTL16 as an m⁶A methyltransferase. The hypomethylated sites were mostly located at coding sequences (CDS) or 3′ untranslated regions (UTR) and enriched near the stop codon, reminiscent of METTL3-regulated m⁶A sites (Fig. 4d, e). Nevertheless, motif analysis revealed that m⁶A peaks regulated by METTL16 are significantly enriched in CAG-containing and/or AG-rich sequences (Fig. 4f), which are different from canonical m⁶A motifs deposited by METTL3 complex (5′-DRACH-3′ or 5′-RRACH-3′, D = A/G/U, R = A/G, H = A/C/U).

To investigate pathways regulated by METTL16-mediated methylation, we performed GO analysis of hypomethylated genes whose expressions were upregulated or downregulated under METTL16 deficiency. Downregulated genes with hypomethylation were enriched in DNA repair and cell cycle process, while upregulated genes were enriched in immune system processes (Fig. 4g, h). Moreover, about half of the genes (118 out of 264 genes) which were downregulated with hypomethylation in erythroblasts from *Mettl16*[fl/fl]*CreERT2*[+] mice were also downregulated in CD71[+]Ter119[+] erythroblasts from *Mettl16*[fl/fl]*Epor-Cre*[+] mice, suggesting that they were similarly regulated by METTL16 in erythroblasts in vivo. The overlapping genes included genes related to DNA repair, such as *Brca1*, *Brca2*, and *Fancm* (Fig. 4i).

METTL16 has been proposed to regulate gene expression by controlling SAM levels via the m⁶A deposition on *MAT2A* mRNA[41,42,55] since MAT2A supplies SAM for METTL3-mediated methylation[56]. To investigate this possibility, we supplemented METTL16-deficient cultured erythroblasts with SAM. Consistent with a previous report[42], SAM treatment reduced *Mat2a* mRNA expression levels in control cells (Supplementary Fig. 9b), indicating that the treatment is sufficient to induce feedback downregulation of *Mat2a* mRNA. However, the SAM treatment failed to restore the expression of DNA-repair-related genes such as *Brca1*, *Brca2*, and *Fancm* under METTL16 deficiency (Supplementary Fig. 9b). Moreover, the expression of DNA-repair-related genes was not greatly affected in human HEL cells[57] lacking the components of the canonical m⁶A writer complex including METTL3 or WTAP, or mouse primary erythroid cells deficient in METTL3 (Supplementary Figs. 9c and 10a, b). Therefore, the mechanism of expression alteration of DNA-damage-associated genes under METTL16 deficiency is not mediated by m⁶A deposition to these mRNA by the METTL3 complex as an indirect effect of METTL16 depletion.

## Identification of METTL16-regulated motifs in DNA-repair-related transcripts

We next explored METTL16-regulated m⁶A motifs in DNA-repair-related transcripts. For this purpose, we focused on two transcripts – *Fancm* and *Brca2*, mRNAs which contain significantly regulated m⁶A peaks in cultured erythroblasts ($p = 2.97 \times 10^{-7}$ or $4.43 \times 10^{-5}$, respectively, Supplementary Data 4, Fig. 5a). Consistently, METTL16 directly interacted with *Fancm* and *Brca2* mRNA to a similar extent to *Mat2a* mRNA or U6 snRNA (Supplementary Fig. 11b). We then confirmed that the expression levels of DNA-repair-related genes including *Fancm* and *Brca2* were downregulated under METTL16 deficiency (Fig. 5b, Supplementary Fig. 11a). Moreover, MeRIP-qPCR experiment confirmed that *Brca2* and *Fancm* mRNAs are less enriched by the lack of METTL16 while *Actb*, *Gata1*, *TfR1*, or *Klf1* transcripts were not particularly enriched (Fig. 5c), suggesting that METTL16 regulates m⁶A deposition on the identified mRNAs specifically. Similar to the METTL16-deficient erythroblasts, the ablation of *Brca2* or *Fancm* gene in cultured erythroblasts reduced *Klf1* mRNA expression, a key transcription factor

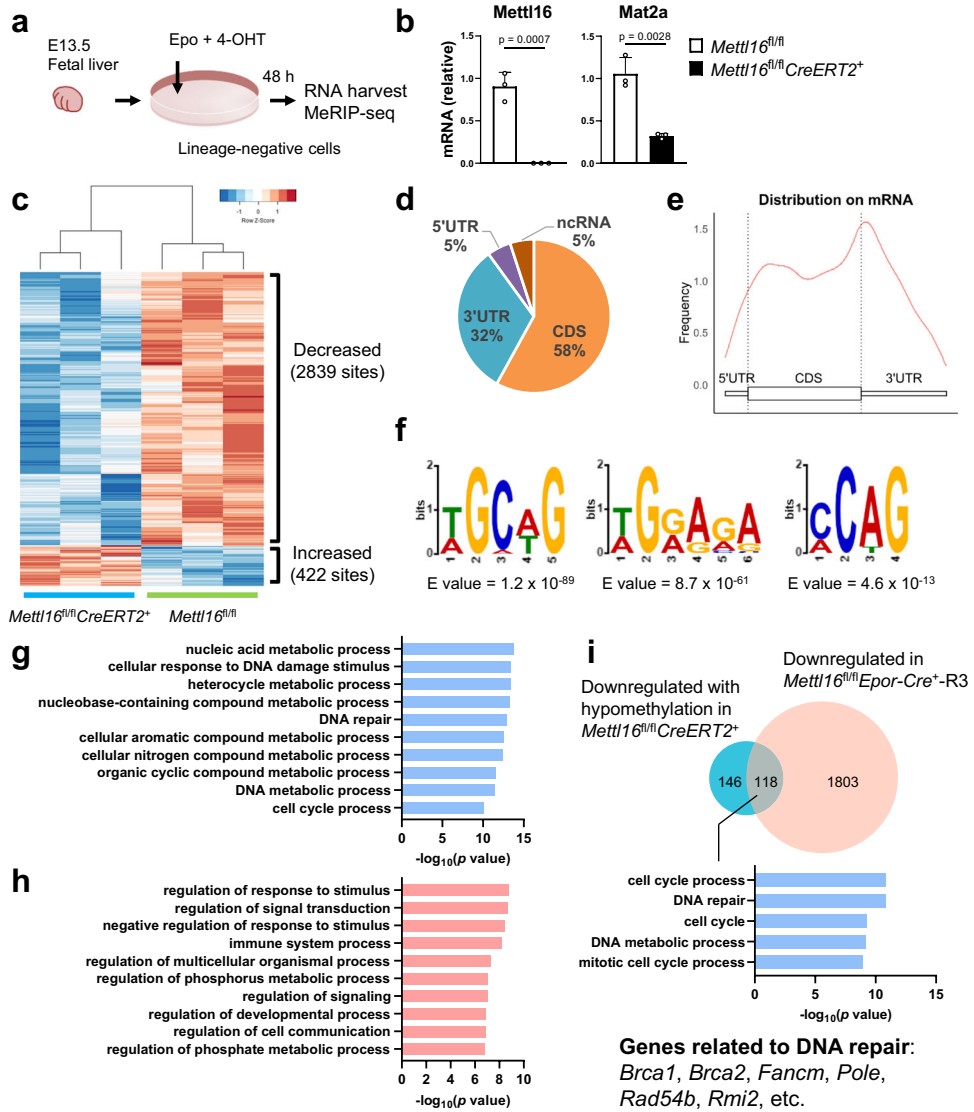

**Fig. 4 | METTL16 regulates the deposition of m⁶A modification on DNA-damage response-associated mRNAs. a** Outline of the in vitro erythroid culture experiments (Methods). **b** mRNA expression levels in cultured erythroblasts from E13.5 control and *Mettl16^fl/fl^CreERT2^+^* mice (n = 3 biological replicates). Data are expressed as mean ± SD. The *p*-values were calculated using two-tailed Student's *t*-test. **c**–**f** MeRIP-seq analyses using poly(A)⁺ RNAs in cultured erythroblasts from control and *Mettl16^fl/fl^CreERT2^+^* mice. Significantly differentially methylated sites were identified using RADAR (n = 3 independent experiments). **c** Heatmap showing the relative methylation levels at these sites. **d** The fractions within the 5′ UTR, CDS, 3′ UTR, and ncRNA. **e** The metagene distribution of the differentially methylated sites in mRNA. **f** Sequence motif logos identified from these sites. **g**, **h** GO analysis of downregulated (**g**) and upregulated (**h**) genes with hypomethylation in erythroblasts from *Mettl16^fl/fl^CreERT2^+^* mice. **i** Venn diagram comparing genes downregulated with hypomethylation in erythroblasts from *Mettl16^fl/fl^CreERT2^+^* mice and genes downregulated in CD71⁺Ter119⁺ erythroblasts from *Mettl16^fl/fl^Epor-Cre^+^* mice. The GO analysis revealed that DNA-repair-related genes were enriched in the intersect. Statistical analysis was performed using Immuno-Navigator[83] (**g**–**i**). Source data are provided as a Source Data file.

for erythroid gene expression (Supplementary Fig. 11c), suggesting that the regulation of these DNA-damage-associated genes by METTL16 is important for proper erythroid differentiation.

U6 snRNA and *MAT2A* mRNA contains METTL16-target consensus sequences 5′-ACAGAGA-3′ (modified A is underlined)[58] and 5′-UACAGAR-3′, respectively[41], that are situated in structured RNA sequences[41,42,55,59]. Therefore, we next asked if transcripts of DNA-repair-related genes also contain such structured sequences. Considering the above-mentioned findings and the enrichment of CAG motifs in our MeRIP-seq analysis (Fig. 4f), we looked for 5′-ACAGAR-3′ boxes in structured motifs. Secondary structure predictions surrounding m⁶A peaks revealed that 5′-ACAGAR-3′ sequences in structured motifs were present near m⁶A peaks in *Fancm* and *Brca2* mRNAs (Fig. 5d). This led us to investigate if m⁶A modifications in these motifs are regulated by METTL16 by utilizing a

method termed SELECT (single-base elongation and ligation based PCR amplification method) to detect m⁶A at a single nucleotide resolution[60]. We observed the reduced methylation levels at the 5′-ACAGAR-3′ sites in *Mat2a*, *Brca2*, and *Fancm* mRNAs, but not in nontarget sites, under METTL16 deficiency (Fig. 5e).

Next, we asked if m⁶A marks regulated by METTL16 control gene expression by generating luciferase reporter vectors, which contain a part of *Brca2* and *Fancm* mRNAs with wild-type or mutated 5′-ACAGAR-3′ boxes (Fig. 5f). The expression levels of the reporters harboring wild-type *Brca2* and *Fancm* sequences were significantly reduced under METTL16 knockdown, while this effect was not observed when the reporter harbored a mutated motif or nontarget sequences including *Klf1* 3′ UTR (Fig. 5g). In summary, these findings indicate that METTL16 directly regulates m⁶A deposition on DNA-repair-related transcripts, thereby controlling gene expression.

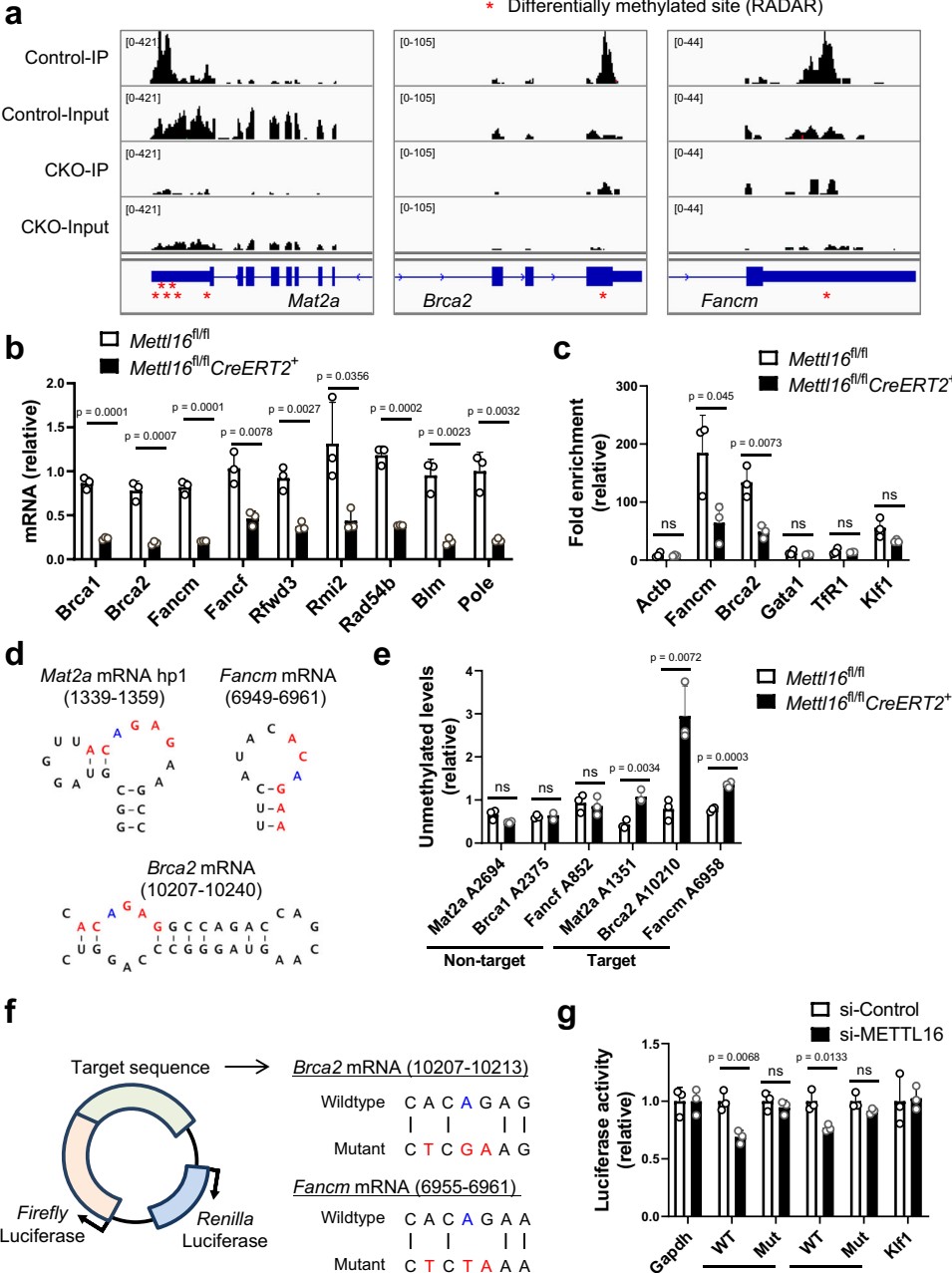

**Fig. 5 | Identification of METTL16-regulated motifs in DNA-repair-related transcripts. a** IGV snapshots of MeRIP-seq reads along indicated mRNAs. Red asterisk indicates differentially methylated sites identified by RADAR. **b, c** mRNA expression levels (**b**) and MeRIP-qPCR validation of m6A methylation levels (**c**) in the E13.5 in vitro cultured erythroblasts from control and *Mettl16*fl/fl*CreERT2*+ (CKO) mice (*n* = 3 biological replicates). **d** Predicted secondary structures surrounding m6A peaks in *Mat2a*, *Brca2*, and *Fancm* mRNA. The ACAGAR boxes were shown in red and the m6A modification sites were shown in blue. **e** SELECT validation of the indicated m6A sites using total RNAs of the E13.5 cultured erythroblasts from control and *Mettl16*fl/fl*CreERT2*+ mice (*n* = 3 biological replicates).

**f, g** Dual-luciferase reporter assay to identify METTL16-regulated transcripts. **f** pmirGLO reporter vector encoding firefly and renilla luciferase were used. Indicated wild-type (WT) and mutant (Mut) sequences were inserted at the 3' end of the firefly luciferase. **g** Luciferase reporter activity of indicated reporters under METTL16 knockdown in HEK293T cells (*n* = 3 biological replicates). Renilla luciferase activity was used as an internal control. Firefly luciferase activity was further normalized to the value of the reporter harboring 3' UTR of *Gapdh* mRNA, which has been reported to have no m6A modification sites[54]. Data are expressed as mean ± SD (**b, c, e, g**). The *p*-values were calculated using two-tailed Student's *t*-test (**b, c, e, g**). ns not significant. Source data are provided as a Source Data file.

## METTL16-mediated mRNA regulation requires MTR4-nuclear RNA exosome complex

It is widely appreciated that epitranscriptomic modifications on mRNAs are recognized by reader RBPs, thereby altering mRNA fate[35,36]. However, our initial CRISPR screen failed to identify canonical m6A reader proteins, including YTHDF and YTHDC family members, as well as other m6A-regulatory RBPs as regulators of TfR1 expression (Supplementary Fig. 12a). This finding led us to explore the regulators

that act downstream of METTL16 by performing a genetic interaction (GI) analysis via pairwise CRISPRi screen[61], which enabled us to predict the functional relationships between two genes systematically. To identify genes whose depletion cancels the effect of METTL16 on TfR1 expression, we prepared the pairwise gRNA library expressing control or METTL16-targeting gRNA, together with gRNAs targeting top 500 candidate regulators (including most of the high confidence hits whose casTLE *p*-value were less than $1 \times 10^{-5}$) of TfR1 expression from

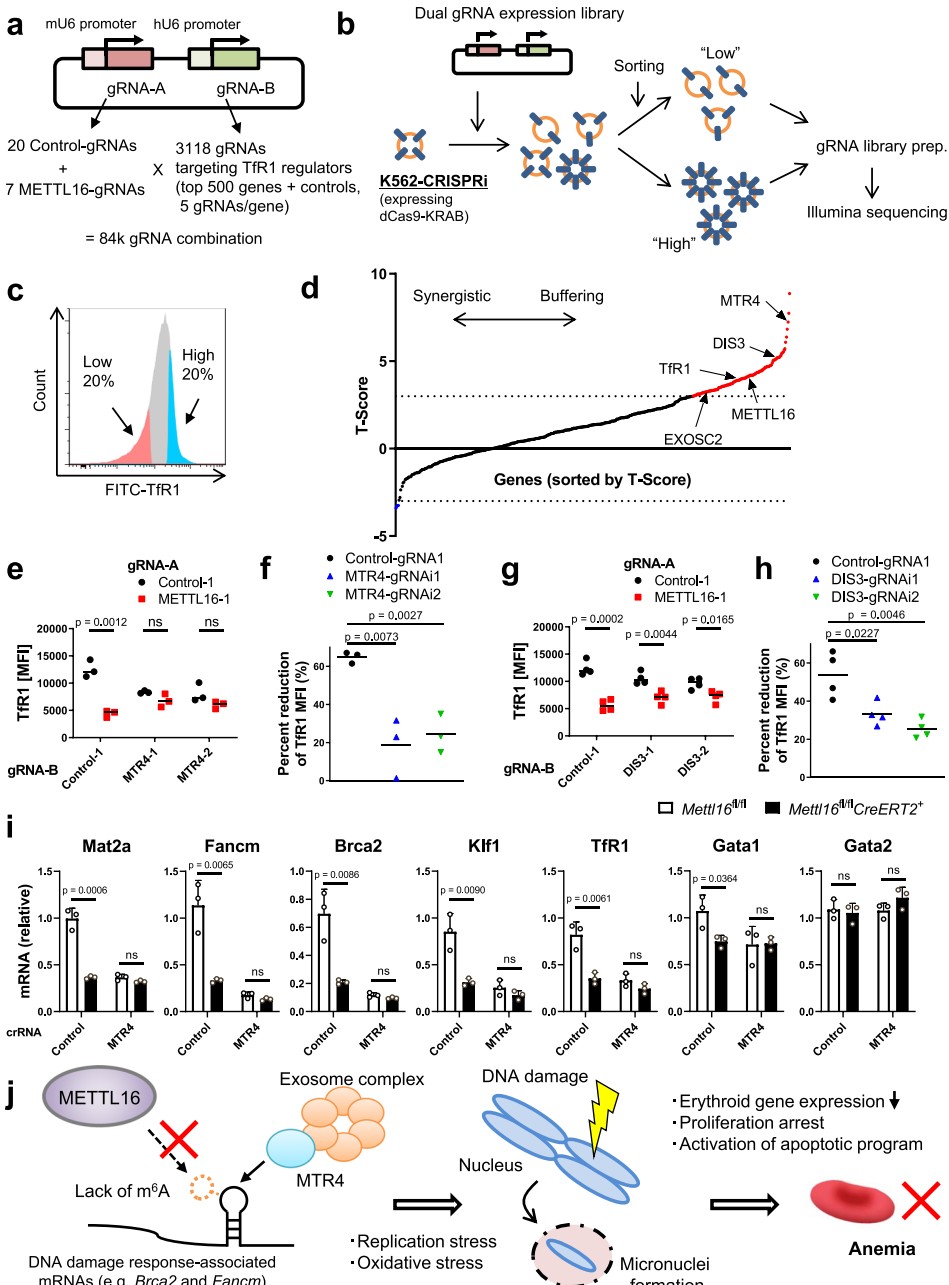

**Fig. 6 | METTL16-mediated mRNA regulation requires MTR4-nuclear RNA exosome complex. a**, **b** Workflow of the pairwise CRISPR screen for genetic interaction analysis. Top 500 TfR1 regulators were selected from the initial genome-wide CRISPR screen results. Then a dual gRNA expression library (M16GI) was constructed (**a**). K562-CRISPRi cells were transduced with the M16GI gRNA library and then cells were sorted according to the surface TfR1 expression (**b**, **c**). **d** Results of the pairwise CRISPR screen. Genes were sorted by T-score, which indicates the degrees of genetic interaction with METTL16. **e**–**h** Validation of the screen hits. Two gRNAs per gene were chosen for validation experiments. K562-CRISPRi cells were transduced with lentiviral vectors encoding individual dual gRNAs. Then cells were treated with DFO overnight, stained with anti-TfR1 antibody and analyzed using flow cytometry (**e**, **g**). Change in TfR1 expression levels [(MFI of METTL16-sufficient cells − MFI of METTL16-deficient cells) / MFI of METTL16-

sufficient cells] in each condition were summarized (**f**, **h**, $n = 3$ biological replicates from independent experiments for MTR4, $n = 4$ for DIS3). Horizontal lines indicate the mean. **i** mRNA expression levels of indicated genes in the cultured erythroblasts from control and *Mettl16*$^{fl/fl}$*CreERT2*$^+$ mice which were nucleofected with indicated crRNAs and Cas9 complexes ($n = 3$ biological replicates from independent experiments). **j** Model overview of METTL16-mediated mRNA regulation in developing erythroid cells, where METTL16 expression is upregulated. METTL16 regulates m⁶A deposition on transcripts of DNA-repair-related genes. The m⁶A deposition regulates the expression of these genes in a manner dependent on the MTR4-nuclear exosome, thereby safeguarding genome integrity and erythropoiesis. Data are expressed as mean ± SD (**i**). The *p*-values were calculated using two-tailed Student's *t*-test (**e**–**i**). ns not significant. Source data are provided as a Source Data file.

the results of our initial CRISPR screen (Methods and Fig. 6a). Then we transduced K562 cells which expressed KRAB-dCas9 (K562-CRISPRi) with this GI library and sorted the cells which expressed high or low levels of TfR1 (Fig. 6b, c). Next, gRNAs in each sorted population were counted, and effect sizes were analyzed as previously described[61]. As a

validation, we analyzed the effect of single gRNAs targeting METTL16 or TfR1 and found that the ablation of these genes indeed downregulates surface TfR1 expression levels (Supplementary Fig. 12b), which was consistent with the results of the initial CRISPR screen. Interestingly, GO terms related to nucleus and RNA processing were

enriched in genes that were required for the effect of METTL16 on TfR1 expression (T-score > 3, Methods, Supplementary Fig. 12c).

Among the top-ranked genes which buffer the METTL16-mediated effect, we found nuclear exosome components, including DIS3 and EXOSC2 (Exosome component 2), and its essential co-factor MTR4 (mRNA transport regulator 4, also known as SKIV2L2), which are central to nuclear RNA degradation and processing[62,63] (Fig. 6d and Supplementary Data 6). Given the high GI score of nuclear exosome components and the critical involvement of exosomes in erythropoiesis[64], we turned to the contribution of this complex in METTL16-mediated mRNA regulation. First, we validated the results of GI analysis by performing individual CRISPRi-knockdowns and confirmed that METTL16 knockdown failed to reduce TfR1 expression in MTR4-depleted cells (Fig. 6e, f). Whereas DIS3 knockdown similarly altered the TfR1 expression under METTL16 knockdown, the effect of DIS3 knockdown was smaller than MTR4 (Fig. 6g, h), potentially due to its functional redundancy with RRP6 (EXOSC10)[65,66]. Next, we asked if MTR4-mediated mechanism is operative in the regulation of METTL16 substrate mRNAs in the cultured erythroblasts. Indeed, MTR4 depletion abrogated the effect of METTL16 ablation on the expression of METTL16 substrate mRNAs such as *Mat2a*, *Fancm*, and *Brca2*, and also on erythroid-important genes, including *Klf1* and *TfR1* (Fig. 6i, Supplementary Fig. 12f). In line with these findings, we found that nuclear-spliced mRNAs were significantly reduced in METTL16-deficient cells, whereas nuclear pre-mRNA (unspliced) levels of *Brca2* and *Fancm* genes were comparable between METTL16-sufficient and -deficient cells (Supplementary Fig. 13a, b). The results support the notion that nuclear mRNA decay is essential for METTL16-mediated mRNA regulation. Moreover, luciferase reporter assays revealed that the MTR4 knockdown abolished the effect of METTL16 on the activity of a reporter harboring the wild-type *Brca2* sequence (Supplementary Fig. 13c). Taken together, the MTR4-nuclear exosome plays a critical role in the regulation of METTL16 substrate mRNAs in erythroblasts.

## Discussion

In the current study, we reveal that METTL16 serves as a critical regulator of erythropoiesis via the maintenance of genome integrity. *Mettl16*-deficient erythroblasts exhibit impaired erythroid differentiation and experience an increased burden of DNA damage in vivo. Mechanistically, METTL16 maintains DNA-damage response-associated mRNAs such as *Brca2* and *Fancm* mRNAs via m[6]A methylation. Furthermore, METTL16 substrate mRNAs are regulated by the MTR4-nuclear exosome in erythroblasts. These findings establish a working model for METTL16-mediated regulation of erythropoiesis (Fig. 6j). During rapid cell proliferation of erythroid cells, increased METTL16 enables the proper expression of DNA-repair-related genes via the m[6]A modification of target mRNAs, where MTR4-nuclear exosome is involved. This leads to the maintenance of genome integrity, the suppression of caspase activation, and presumably the protection of GATA-1, which facilitates the establishment of erythroid identity and differentiation. Together with a previous study showing the critical role of the canonical m[6]A writer complex in human erythroid cell differentiation[57], our findings highlight the importance of RNA m[6]A methylation in the regulation of erythroid differentiation.

Although METTL16 knockout embryos showed early lethality[55], the generation of conditional knockout mice enabled us to uncover its essential role in erythroid differentiation. Further, it appears that the proper maintenance of genome integrity by METTL16 is critically required for the development of rapidly dividing cells like erythroblasts. It would be interesting to investigate in future studies if METTL16 governs DNA-damage responses in a variety of tissues/cells, and if there are cell type-specific functions via modifying distinct target mRNAs.

GATA-1 is an essential molecule for erythroid differentiation as exemplified by Diamond-Blackfan anemia, where the specific

downregulation of GATA-1 is involved[67]. METTL16 seems to promote erythropoiesis indirectly via the regulation of GATA-1 expression, thereby upregulating erythroid-important genes such as TfR1 and KLF1[9,68]. In support of this notion, we found the downregulation of GATA-1 and upregulation of cleaved caspase-3 (Fig. 3f, Supplementary Fig. 5a, b), which has been implicated in the degradation of GATA-1 in erythroblasts[50–52]. Moreover, DNA-damage-associated genes were greatly downregulated by the lack of METTL16, which can trigger the activation of caspase-3 (Fig. 5b, Supplementary Data 5). Furthermore, the depletion of *Brca2* or *Fancm* gene in cultured erythroblasts reduced *Klf1* mRNA expression, similar to METTL16 deficiency (Supplementary Fig. 11c). These findings imply that GATA-1 protein expression is tightly controlled by the balance between METTL16-DNA-repair machinery and DNA-damage-caspase axis to regulate the downstream erythroid differentiation program.

The epitranscriptome mark m[6]A is deposited by several methyltransferases[36]. While METTL3 and METTL14 broadly install m[6]A in mRNAs and ncRNAs and exerts diverse functions[35,69], METTL16 was known to deposit m[6]A on only a handful of substrate RNAs such as *MAT2A* mRNA and U6 snRNA[41,42,55]. In this study, MeRIP-seq analysis revealed that METTL16 regulates m[6]A deposition on the transcripts of DNA-damage-associated genes, including *Brca2* and *Fancm* mRNAs, which contribute to the maintenance of genome integrity. Since motifs for the m[6]A deposition by METTL16 are distinct from those by the METTL3 and METTL14 complex, it is tempting to speculate that METTL16 serves as a more specialized m[6]A writer than METTL3 for limited functions such as the maintenance of intracellular metabolite levels and genome integrity.

We found that METTL16 regulates m[6]A deposition on *Brca2* and *Fancm* mRNAs, and the MTR4-nuclear exosome is required for the control of these mRNAs. Although our findings strongly support the genetic interaction between METTL16 and MTR4-nuclear exosome, an open question is if and how METTL16-mediated modification is recognized or discriminated by the MTR4-nuclear exosome. The m[6]A deposition may alter the secondary and/or tertiary structures of RNAs, leading to their protection from the ribonucleolytic decay by the MTR4-nuclear exosome, since the chemical nature of m[6]A has been associated with the regulation of secondary structures. From an m[6]A thermodynamic perspective, *syn* conformation is energetically favored over *anti* conformation[70]. Nevertheless, the structural studies suggest that the m[6]A-U base pairing in RNA duplex takes unfavorable *anti* conformation. Consistently, a thermodynamic study demonstrated that m[6]A destabilizes A-U base pairs, but stabilizes A–A pairs[70], as experimentally shown in another study as well[43]. Moreover, it has been revealed that m[6]A stabilizes RNA duplex if m[6]A is located at the overhanging positions[70]. Another biochemical study has also shown that m[6]A modification alters local structures, thereby controlling the accessibility of indirect reader proteins[71]. Yet-identified RNA-binding protein(s) as readers for m[6]A and/or secondary structures may be involved to mediate the recognition by the MTR4-nuclear exosome. Taken together, the structural change by m[6]A deposition may underlie the mechanism that regulates METTL16-target mRNAs, which warrants further studies.

By performing a genome-wide CRISPR screen, we identified multiple regulators and especially highlighted the importance of post-transcriptional regulation in erythroid gene expression as represented by TfR1 regulation. Besides METTL16 and IRP2, we isolated many RBPs previously uncharacterized as regulators of erythropoiesis and/or iron homeostasis. These RBPs include HNRNPL and XPO5, and the pairwise CRISPRi screen revealed that they can function in a manner distinct from METTL16 (Supplementary Data 1 and 6). Therefore, the RBPs regulate the expression of erythroid genes via different steps of mRNA metabolism. Further investigation of these RBPs will provide a comprehensive picture of mRNA regulatory network in erythropoiesis.

We identified the structured motifs targeted by METTL16 on murine *Brca2* and *Fancm* mRNAs. However, it appears that these motifs are not conserved in human genes. It is of note that the sequence conservation is not always seen in METTL16 targets as in the case of SAM-producing enzymes (*MAT2A* in mammals and *sams* in worms). Mammalian and worm METTL16 commonly regulate SAM-producing enzymes in response to SAM concentrations without the conservation of METTL16-target sites and regulatory mechanisms[41,42,44,72]. Since it has been known that murine and human methylome is substantially distinct[73], functional conservation of m⁶A-mediated regulation, rather than sequences themselves, may be prevalent. It would be interesting to investigate DNA-damage-associated METTL16 targets and their regulation in human cells in future studies.

The dysregulation of epitranscriptome has been implicated in the pathogenesis of various human diseases such as cancer and genetic disorders[74,75]. Given that METTL16 is pivotal for the regulation of DNA-damage response and erythropoiesis, the altered METTL16 expression and/or activity may also be associated with diseases such as hematological malignancies. Future studies will uncover the role of METTL16 and the therapeutic value targeting the expression or enzymatic activity of METTL16 in such pathogenic conditions.

## Methods

### Mice

EpoR-Cre and ROSA26-CreERT2 mice have been previously described[76,77]. *Tmem173*^gt/gt mice were kindly provided by Russell Vance and Takashi Fujita[78]. *Ifnar1*^−/− mice were obtained from B&K Universal. *Mettl16*-floxed mice were generated using the two-step CRISPR/Cas9-mediated genome editing method as described previously[79] (Supplementary Fig. 2a). Briefly, Cas9-gRNA ribonucleoprotein complex and donor oligos for each loxP sequence were introduced by electroporation at 1-cell and 2-cell stage, respectively. The gRNAs and donor oligos used for the generation of the *Mettl16*-floxed allele are listed in Supplementary Table 4. The targeted locus was PCR amplified from the tail genome of pups, cloned into a TA or TOPO vector using the TA Cloning Kit (Invitrogen) or Zero Blunt TOPO PCR Cloning Kit (Invitrogen), respectively, and sequenced to confirm successful genome editing. *Mettl16*-floxed mice were crossed with EpoR-Cre and ROSA26-CreERT2 mice to obtain conditional knockout mice. Unless otherwise specified, wild-type, heterozygous or Cre-negative mice were used as controls since heterozygous mice did not show any detectable phenotype. All animal experiments were conducted in compliance with the regulations approved by the Committee for Animal Experiments of the Institute for Frontier Life and Medical Sciences and Graduate School of Medicine, Kyoto University. Mice were maintained at the animal facilities with a 12 h light/dark cycle and access to food and water ad libitum. Room temperature was maintained at $23 \pm 3 \,°C$, with a relative humidity of $50 \pm 20\%$.

Timed embryos were obtained by natural mating or in vitro fertilization (IVF). For natural mating, male and female mice were bred overnight, and the presence of the vaginal plug was checked the next morning. The pregnant mice were sacrificed via cervical dislocation at the indicated time after mating, and the fetuses were harvested. Images of fetuses were acquired using the stereoscopic microscope SZX16 (Olympus). Fetuses were immediately decapitated, and the fetal livers were harvested for further analysis.

### Cell lines

HEK293T (#CRL-3216), K562 (#CCL-243), HeLa (#CCL-2), and NIH3T3 (#CRL-1658) were obtained from ATCC. All cell lines were confirmed to be mycoplasma negative by using PCR Mycoplasma Detection Kit (Applied Biological Materials). K562 cell lines that stably express wild-type Cas9 endonuclease or nuclease-dead Cas9 fused with a KRAB domain (K562-CRISPRi) were generated as described previously[29,80]. HEK293T, NIH3T3, and HeLa cells were cultured in Dulbecco's modified eagle medium (DMEM) supplemented with 10% FBS, 100 U/mL of penicillin, 100 μg/mL of streptomycin, and 50 μM 2-mercaptoethanol. K562 cell lines were cultured in RPMI-1640 medium supplemented with 10% FBS, 100 U/mL of penicillin, and 100 μg/mL of streptomycin.

Cells were transfected using Lipofectamine 2000, LTX, RNAiMAX (Invitrogen), PEI "Max" (Polysciences), Neon transfection system (Thermo Fisher Scientific) or 4D-Nucleofector Core and X Unit and P3 Primary Cell 4D-Nucleofector X Kit (Lonza) according to the manufacturer's instructions.

For the erythroid differentiation of K562 cells, cells were treated with 50 μM hemin (Sigma–Aldrich) for 24 h.

### Reagents

Primary antibodies for immunoblot and immunofluorescence analysis used in this study were as follows: anti-METTL16 (Bethyl Laboratorties, #A304-192A, 1:1000), anti-transferrin receptor (Santa Cruz Biotechnology, #sc-32272, 1:1000), anti-β actin (Santa Cruz Biotechnology, #sc-1615, 1:5000), anti-Phospho-Histone H2A.X (Ser139, Cell Signaling Technology, 2577, 1:800), anti-cleaved caspase-3 (Cell Signaling Technology, 9664, 1:2000), anti-GATA-1 (Cell Signaling Technology, 3535, 1:500), anti-FASN (Santa Cruz Biotechnology, #sc-55580, 1:1000), anti-XPO5 (Santa Cruz Biotechnology, #sc-271036, 1:1000), anti-HNRNPL (Santa Cruz Biotechnology, #sc-32317, 1:1000), anti-MTR4 (Santa Cruz Biotechnology, #sc-515828, 1:1000), anti-METTL3 (Proteintech, #15073-1-AP, 1:1000), anti-BRCA2 (Abcam, #ab27976, 1:1000), anti-FANCM (Bethyl Laboratorties, #A302-637A, 1:1000), anti-GAPDH (Santa Cruz Biotechnology, #sc-47724, 1:1000), anti-SFPQ (MBL, #RN014MW, 1:1000), and anti-FLAG (Sigma–Aldrich, #F3165, 1:5000). Secondary HRP-conjugated antibodies were from Cytiva (Anti-Mouse IgG, HRP-Linked F(ab')2 Fragment Sheep, #NA9310, and Anti-Rabbit IgG, HRP-Linked F(ab')2 Fragment Donkey, #NA9340, 1:5000).

Antibodies and fluorescent dyes for flow cytometry analysis were as follows: Brilliant Violet 421 Rat anti-mouse CD16/32 (Clone: 93, Biolegend, #101332, 1:200), FITC Mouse anti-human CD71 (Clone: CY1G4, Biolegend, #334104, 1:100), FITC Rat anti-mouse CD34 (Clone: RAM34, BD Pharmingen, #560238, 1:100), PerCP/Cy5.5 Streptavidin (Biolegend, #405214, 1:500), PE Rat anti-mouse Ter119 (Clone: TER-119, Biolegend, #116207, 1:200), PE Rat anti-mouse c-kit (Clone: 2B8, Biolegend, #105808, 1:200), PE/Cy7 Rat anti-mouse Ki-67 (Clone: 16A8, Biolegend, #652425, 1:200), PE/Cy7 Annexin V (Biolegend, #640949, 1:200), PE/Cy7 Rat anti-mouse Sca-1 (Clone: D7, Biolegend, #108114, 1:200), APC Mouse anti-human CD71 (Clone: CY1G4, Biolegend, #334107, 1:200), and APC Rat anti-mouse CD71 (Clone: RI7217, Biolegend, #113819, 1:200).

Deferoxamine mesylate (DFO) and cycloleucine were purchased from Sigma–Aldrich. S-(5′-Adenosyl)-L-methionine (sulfate tosylate) was purchased from Cayman Chemical. Media for cell culture were from Nacalai Tesque unless otherwise specified. siRNAs (Silencer Select) were obtained from Invitrogen. Silencer Select Negative Control No.1 siRNA (Invitrogen) was used as a negative control. Alt-R CRISPR-Cas9 crRNA-XTs and Alt-R CRISPR-Cas9 tracrRNA were obtained from Integrated DNA Technologies. siRNAs and crRNAs used in this study are listed in Supplementary Tables 1 and 5.

### Plasmids

Lentiviral packaging components were described previously[29]. Individual gRNAs for individual knockouts were inserted into the pMCB320 vector (Addgene #89359). The sequences of individual gRNAs are listed in Supplementary Table 1. pInducer20 (no ccdB) was generated from pInducer20 (Addgene #44012) and pENTR1A no ccDB (w48-1) (Addgene #17398) using Gateway LR Clonase Enzyme Mix (Thermo Fisher Scientific). To obtain pInducer20-BSD, IRES2-BSD was amplified from CSII-CMV-MCS-IRES2-Bsd (#RDB04385) and cloned into the

pInducer20 (no ccdB) at the 3′ end of the rtTA-Advanced coding sequence. A full-length coding sequence of human METTL16 was inserted into the pInducer20-BSD or pFLAG-CMV2 (Sigma–Aldrich) vector. The gRNA-resistant wild-type, PP185/186AA and F187G mutants of METTL16 coding sequence were generated using QuikChange Lightning Site-Directed Mutagenesis Kit (Agilent). pmirGLO was purchased from Promega. The respective parts of mouse *Brca2*, *Fancm*, *Gapdh*, and *Klf1* mRNA was inserted into pmirGLO. Mutant sequences of mouse *Brca2* and *Fancm* mRNA were generated using QuikChange Lightning Site-Directed Mutagenesis Kit (Agilent).

To generate dual gRNA-expressing vectors, pMCB320 vector were digested with BstXI and BlpI and ligated with synthesized annealed oligos (Eurofins Genomics). The pKHH030 vector (Addgene #89358) was digested with BbsI and ligated with annealed oligos to obtain hU6 cassette containing controls or METTL16-targeting gRNAs, individually. The resultant pKHH030 vectors were digested with BamHI and XhoI to obtain the hU6 cassette. The hU6 cassettes were ligated into pMCB320 vectors, which contain single gRNAs.

### Lentivirus production and infection

Lentivirus was produced according to the method described previously[29]. Briefly, HEK293T cells were transfected with lentiviral packaging components (pMDL, pRSV and pMD2) together with a transfer vector using Polyethylenimine "Max" (Polysciences). The transfected cells were incubated for 72 h. The culture supernatant was collected, filtered through 0.45 μm filter (Merck), and used for transduction. Then Cas9-expressing K562 or K562-CRISPRi cell line was harvested, resuspended in viral supernatant supplemented with polybrene (8.0 μg/mL, Sigma–Aldrich), and spin-infected for 3 h. The cells were washed in DPBS and seeded in a fresh medium. After 2 days of transduction, the cells were harvested and analyzed by flow cytometry to check the transduction efficiency. Then puromycin was added to the medium and incubated for 3–5 days for the selection of gRNA-expressing cells.

### Genome-wide CRISPR/Cas9 screen

The CRISPR/Cas9 gRNA lentiviral library (containing 10 gRNAs/gene and negative control gRNAs) was designed and generated as described previously[31]. The total gRNA library was divided into nine sublibraries (Addgene #101926-34). The samples for each sublibrary were transduced and prepared separately throughout the experiment. Lentiviral packaging was performed as described above. The Cas9-expressing K562 cells were transduced with lentiviral sublibraries and maintained for 2 days without puromycin. Then the transduction efficiency was confirmed by performing flow cytometric analysis (~30% mCherry positive cells to ensure the single viral transduction), followed by the selection with puromycin for at least 5 days. Throughout the culture, the cell numbers equal to at least 1000-fold representation of sublibraries were maintained. Then the gRNA-transduced cells were harvested, stained with FITC anti-TfR1 (CD71) antibody (BioLegend), washed, and resuspended in MACS buffer (0.5% bovine serum albumin (Sigma–Aldrich), 2 mM EDTA and 1× DPBS). Cell sorting was performed using SH800Z cell sorter (Sony). One hundred micrometers sorting tip (Sony) was used for cell sorting. Cells that express the highest or lowest TfR1 levels (~10% each) were sorted as shown in Fig. 1b based on the expression of mCherry and CD71 (TfR1). In each experiment, cell numbers equal to at least 100-fold library representation were collected for each gated population. Then sorted cells were cryopreserved before cells were subjected to DNA isolation.

### DNA isolation and library preparation for genome-wide CRISPR screen

DNA isolation was performed using QIAamp DNA Blood Midi/Maxi Kit according to the manufacturer's instructions (QIAGEN). The library preparation was performed as described previously[29]. Briefly, isolated DNA was used to amplify the region which contained gRNA sequences using the Herculase II Fusion DNA Polymerase (Agilent). Next, a nested PCR was performed to add Illumina adapter sequences and barcodes for each sample. The PCR products were electrophoresed in TBE/Agarose gel. The specific bands were excised and purified using QIAquick Gel Extraction Kit (QIAGEN) and concentrated using QIAquick PCR Purification Kit (QIAGEN). The DNA concentration of the prepared libraries was determined using NEBNext Library Quant Kit for Illumina according to the manufacturer's instructions (New England BioLabs). The libraries were pooled and diluted in EB buffer (QIAGEN). The concentration of the diluted and pooled libraries was again determined using NEBNext Library Quant Kit for Illumina. Next, the libraries were denatured and further diluted according to the protocol of Illumina.

### Analysis of screen results

Sequencing was performed using NextSeq 500 Sequencer and NextSeq 500/550 High-Output v2 Kit (Illumina) using custom primers as described previously[29]. The casTLE package was used under the Python environment to analyze the reads[32]. To control the false discovery rate (FDR), q-value was calculated using the Benjamini-Hochberg procedure. The screen results were visualized by using casTLE.

### Construction of METTL16 genetic interaction library

The focused METTL16 genetic interaction library (M16GI library) was constructed as described previously with slight modifications[61]. Five CRISPRi gRNAs/genes for the top 500 TfR1 regulators from the previously described gRNA list[81] were selected. gRNAs which contained BamHI and XhoI sites were excluded to avoid digestion during cloning. We also added negative control gRNAs to the list. gRNA oligo pools were generated and obtained from CustomArray. The oligos were amplified by PCR, digested with BstXI and BlpI, and ligated into pMCB320 vectors (Addgene #89359), which contained mU6-gRNA-tracrRNA cassette.

Next, a hU6 gRNA-tracrRNA cassette was obtained using a different cloning method. The pKHH030 vector (Addgene #89358) was digested with BbsI and ligated with annealed oligos to obtain the hU6 cassette containing 20 controls or seven METTL16-targeting gRNAs, individually. The resultant pKHH030 vectors were combined and digested with BamHI and XhoI to obtain the hU6 cassette. The hU6 cassettes were ligated into pMCB320 vectors, which contained gRNAs for TfR1 regulators. In all, the cloning generated 84k dual gRNA combinations.

### CRISPR/Cas9 screen with M16GI library

K562-CRISPRi was transduced with the M16GI library, and 2 days after transduction, cells were selected with puromycin for 3 days. Then cells were resuspended in the media without puromycin and incubated overnight. Next, cells were treated with DFO (100 μM) overnight, stained with FITC anti-TfR1 antibody (BioLegend), and sorted as shown in Fig. 6c.

### DNA isolation and library preparation for genetic interaction screen

DNA isolation was performed using QIAamp DNA Blood Midi/Maxi Kit according to the manufacturer's instructions (QIAGEN). The library preparation was performed using a newly developed direct PCR protocol. This protocol is an improved method compared to the original library preparation method, which was described previously[61]. First, genomic DNA was amplified using barcoded primer pairs and Herculase II Fusion DNA Polymerase (Agilent). The PCR products were mixed and then electrophoresed in TBE/Agarose gel. The specific bands were excised and purified using QIAquick Gel Extraction Kit (QIAGEN) and concentrated using QIAquick PCR Purification Kit (QIAGEN).

## Bioinformatics analysis

Measurement of genetic interactions has been performed as described previously[61]. Gene ontology (GO) analysis was performed using Immuno-Navigator[82]. The output from Immuno-Navigator was visualized using REVIGO[83]. GSEA was performed using GSEA software (Broad Institute and University of California, https://www.gsea-msigdb.org/gsea/).

## Flow cytometric analysis

Cells were stained with Zombie NIR Fixable Viability Kit (Biolegend) or Fixable Viability Dye eFluor 780 (Thermo Fisher Scientific), followed by the staining with fluorescent-labeled antibodies for 15 min. The cells were then washed and resuspended in MACS buffer (0.5% bovine serum albumin (Sigma–Aldrich), 2 mM EDTA and 1× DPBS). For intracellular staining, cells were fixed and permeabilized using the Foxp3/ Transcription Factor Staining Buffer Set (eBioscience). Flow cytometric data were collected using FACSVerse or LSRFortessa X-20 (BD Biosciences) or SH800Z (Sony). Collected data were analyzed with FlowJo (BD Biosciences).

## Immunoblot analysis

Samples were prepared in RIPA buffer (20 mM Tris-HCl (pH 8.0), 150 mM NaCl, 10 mM EDTA (pH 8.0), 1% NP-40 (Nacalai Tesque), 1% SDS (Nacalai Tesque), 1% sodium deoxycholate (Sigma–Aldrich), supplemented with cOmplete, Mini (Roche)) or RIP Lysis buffer (20 mM Tris-HCl (pH 7.4), 100 mM KCl, 1.5 mM MgCl$_2$, 0.5% NP-40, supplemented with cOmplete, Mini (Roche)). Samples were mixed with loading buffer containing β-mercaptoethanol, boiled at 95 °C for 5 min, and cooled on ice. Samples were loaded on 5–20% polyacrylamide gels (ATTO), electrophoresed, and transferred to PVDF membranes (Bio-Rad). For BRCA2 and FANCM proteins, samples were loaded on NuPAGE 3–8% Tris-Acetate Protein Gels (Thermo Fisher Scientific). The membranes were immersed in 5% skim milk (BD Difco) in TBS-T. Signal Enhancer HIKARI for Western Blotting and ELISA (Nacalai Tesque) or 5% skim milk in TBS-T was used to dilute primary and secondary antibodies. Blots were developed using Luminata Forte Western HRP Substrate (Millipore) according to the manufacturer's instructions. Chemiluminescent detection was performed using Amersham Imager 600 (GE Healthcare).

## Immunostaining

Sorted CD71$^+$Ter119$^+$ fetal liver erythroblasts from E11.5 embryos were attached to a slide glass using Shandon Cytospin 4 (Thermo Fisher Scientific). Cells were fixed with 4% paraformaldehyde phosphate buffer solution (Nacalai Tesque) for 15 min, permeabilized with 0.5% Triton X-100 and 0.1% gelatin in PBS for 10 min and blocked with 0.1% gelatin and 2% goat-serum (#143-06561; FUJIFILM Wako Pure Chemical) in PBS for 30 min at room temperature. Then cells were incubated with primary antibody against γ-H2AX (Ser139, #2577, Cell Signaling Technology) or GATA-1 (#3535, Cell Signaling Technology), followed by the incubation with secondary antibody conjugated with Alexa Fluor 568 (Thermo Fisher Scientific) and Hoechst 33342 (Thermo Fisher Scientific). After washing, samples were mounted using ProLong Diamond Antifade Mountant (Thermo Fisher Scientific) and air-dried. Images were acquired using TCS SPE (Leica) and analyzed using ImageJ (NIH). The presence of micronuclei and γ-H2AX was determined manually by two independent investigators.

## RNA sequencing (RNA-seq)

Fetal liver erythroblasts from E11.5 embryos were stained with Zombie NIR Fixable Viability Kit (Biolegend) to discriminate dead cells, followed by the staining with anti-CD71 and anti-Ter119 antibodies (Biolegend). Live CD71$^+$Ter119$^+$ and CD71$^+$Ter119$^-$ cells were sorted using an SH800Z cell sorter. The total RNA of sorted cells was isolated using NucleoSpin RNA XS (Takara Bio) according to the manufacturer's protocol.

RNA integrity (RIN) was examined using Bioanalyzer (Agilent) with RNA 6000 Pico Kit (Agilent) to confirm the RIN > 7. Then, the RNA-seq libraries for Illumina sequencing were generated using NEBNext Ultra II Directional RNA Library Prep Kit for Illumina (New England BioLabs) according to the manufacturer's protocol. Sequencing was performed using NextSeq 500 Sequencer and NextSeq 500/550 High-Output v2 Kit (Illumina).

The RNA-seq analysis was performed using the Galaxy web server (use.galaxy.org). Raw reads were trimmed by cutadapt (http://journal.embnet.org/index.php/embnetjournal/article/view/200), aligned to the mm10 murine reference genome by HISAT2[84], and then counted via the featureCounts package[85]. The read counts were further processed by limma[86].

## Isoform switch analysis

Alternative splicing and isoform switch events were analyzed using IsoformSwitchAnalyzeR[53]. Briefly, RNA-seq datasets were trimmed using cutadapt. Then salmon was used to quantify the expression of transcript isoforms[87]. Isoform switch events were analyzed using DEXseq[88] via IsoformSwitchAnalyzeR. Then transcript annotations were generated using CPAT (http://lilab.research.bcm.edu/), Pfam (https://www.ebi.ac.uk/interpro/), and SignalP (https://services.healthtech.dtu.dk/service.php?SignalP-5.0), and open reading frames (ORFs) and premature termination codons (PTCs) were predicted via IsoformSwitchAnalyzeR.

## Quantitative PCR (qPCR) analysis

TRIzol (Invitrogen) was used for the isolation of total RNA, and ReverTra Ace with gDNA Remover (Toyobo) was used for cDNA synthesis according to the manufacturer's instructions. For quantitative PCR, the synthesized cDNAs were amplified using PowerUp SYBR Green Master Mix (Applied Biosystems) according to the manufacturer's instructions. Fluorescence was detected using a 7500 real-time PCR system (Applied Biosystems). To determine the relative expression, mRNA expression levels of genes were normalized to the expression level of glyceraldehyde 3-phosphate dehydrogenase (*Gapdh*) or β-actin (*Actb*) unless otherwise specified. Primers used for qPCR analysis are listed in Supplementary Table 2.

## Methylated RNA immunoprecipitation sequencing (MeRIP-seq)

MeRIP-seq was performed as described previously with slight modifications[54]. Total RNA from cultured fetal liver cells was isolated using TRIzol followed by RNeasy Mini Kit with RNase-Free DNase Set (QIAGEN). Poly(A)$^+$ mRNA was isolated from 15 μg of total RNA using NEBNext Poly(A) mRNA Magnetic Isolation Module (New England Biolabs). mRNAs were fragmented using RNA Fragmentation Reagents (Thermo Fisher Scientific) at 75 °C for 12 min and analyzed using Bioanalyzer RNA 6000 Pico Kit (Agilent). 1/15 of the fragmented mRNA was removed and used for input. The rest of the samples was used for m$^6$A-immunoprecipitation (IP) using EpiMark $N^6$-Methyladenosine Enrichment Kit (New England Biolabs) according to the manufacturer's protocol. Next, the MeRIP-seq library was prepared using NEBNext Ultra™ II Directional RNA Library Prep Kit for Illumina (New England Biolabs). First and second-strand cDNA synthesis was performed according to the manufacturer's protocol. The cDNA was purified using DNA Clean & Concentrator-5 (Zymo Research). Next, end-prep of cDNAs, adaptor ligation, and USER enzyme treatment was performed according to the manufacturer's protocol. Then, the ligation reactions were purified using Sample Purification Beads (1.2x). Next, PCR enrichment of the adaptor ligated DNA was performed according to the manufacturer's protocol (11 cycles for input, and 14 cycles for IP samples). Then, the PCR products were purified using Sample Purification Beads (1.2×) twice. Library size and concentration were determined using Bioanalyzer High Sensitivity DNA Kit (Agilent) and NEBNext Library Quant Kit for Illumina (New England Biolabs).

Sequencing was performed using the NextSeq 500 Sequencer and NextSeq 500/550 High-Output v2 Kit (Illumina).

## Analysis of MeRIP-seq data

After demultiplexing, raw sequencing reads were trimmed using Trim Galore (https://github.com/FelixKrueger/TrimGalore). Trimmed reads were mapped to the mouse genome using HISAT2 and the reads were counted via the featureCounts package under the R environment (v3.4.0)[85]. To analyze differential methylation sites between control and *Mettl16*-deficient erythroblasts while removing batch effects, the pipeline RADAR (v0.2.1) was used[89]. For motif discovery, each peak was extended 50 nt upstream of its 5′ end and downstream of its 3′ end as described in RADAR Manual (https://scottzijiezhang.github.io/RADARmanual/) and analyzed by MEME (v5.4.1)[90] with the arguments -mod zoops -minw 4 -maxw 6 -objfun de -markov_order 0. Control sequences were generated by shuffling the primary sequences.

## MeRIP-qPCR analysis

Total RNA from cultured fetal liver cells was isolated using TRIzol followed by RNeasy Mini Kit with RNase-Free DNase Set (QIAGEN). Then poly(A)$^+$ mRNA was isolated from 3 to 5 µg of total RNA using NEBNext Poly(A) mRNA Magnetic Isolation Module (New England Biolabs). Then the isolated RNAs were denatured at 65 °C for 5 min, cooled on ice, and used for m$^6$A-IP using EpiMark $N^6$-Methyladenosine Enrichment Kit (New England Biolabs) according to the manufacturer's protocol. IP and input samples were used for cDNA synthesis and qPCR analysis as described above. mRNA expression levels of genes were normalized to *Gapdh*.

## Single-base elongation and ligation-based qPCR amplification (SELECT)

SELECT analysis was performed as described previously[60]. Briefly, total RNA was isolated using TRIzol followed by RNeasy Mini Kit (QIAGEN) with RNase-Free DNase Set (QIAGEN). Then an equal amount of RNA samples was mixed with 40 nM up and down probes and 5 µM dNTP in 1× CutSmart Buffer (New England Biolabs). Samples were denatured at 90 °C for 1 min, and then gradually cooled to 40 °C. Next, 0.01 U Bst 2.0 polymerase (New England Biolabs), 0.5 U SplintR ligase (New England Biolabs), and 10 nM ATP (New England Biolabs) were added to the reaction mixture. Samples were incubated at 40 °C for 20 min, then at 80 °C for 20 min, and cooled to 4 °C. The final reaction mixtures were used for qPCR analysis as described above. Primers used for SELECT analysis are listed in Supplementary Table 3. METTL16 target and nontarget sites were chosen based on the MeRIP-seq results.

## RNA immunoprecipitation

RNA Immunoprecipitation was performed as previously described with slight modifications[12]. NIH3T3 cells were transfected with the expression vector for wild-type METTL16, or empty plasmid as a control. After 24 h of incubation, the medium was removed, and cells were lysed in RIP Lysis buffer (20 mM Tris-HCl (pH 7.4), 100 mM KCl, 1.5 mM MgCl$_2$, 0.5% NP-40, 0.2 U/mL RNasin (Promega) and Complete Mini Protease Inhibitor Cocktail). Lysates were aliquoted to obtain input samples. Then cell lysates were mixed with Dynabeads Protein G (Invitrogen) preincubated with anti-FLAG antibody, and incubated at 4 °C for 3 h with gentle rotation. Then the beads were immobilized, and washed five times in RIP Wash buffer (20 mM Tris-HCl (pH 7.4), 150 mM KCl, 2.5 mM MgCl$_2$, 0.2 U/mL RNasin, and Complete Mini Protease Inhibitor Cocktail), and mixed with TRIzol (Invitrogen).

## Histological and cytological analysis

Tissue samples were fixed in 10 N Mildform (FUJIFILM Wako Pure Chemical). Tissue sections were prepared in the Center for Anatomical, Pathological, and Forensic Medical Research, Kyoto University Graduate School of Medicine.

Fetal liver cell slides were prepared using Shandon Cytospin 4 (Thermo Fisher Scientific). The cell slides were stained with standard May-Grunwald and Giemsa solutions (FUJIFILM Wako Pure Chemical and Sigma−Aldrich, respectively).

## Immunohistochemical staining

After deparaffinization and antigen retrieval using the microwave, endogenous peroxidase activity was blocked by 0.3% H$_2$O$_2$ in methyl alcohol for 30 min. The glass slides were washed in PBS (6 times, 5 min each) and mounted with 1% normal serum in PBS for 30 min. Subsequently, the primary antibody (anti-cleaved caspase-3, Cell Signaling Technology) was applied overnight at 4 °C. The slides were then incubated with a biotinylated secondary antibody diluted to 1:300 in PBS for 40 min, followed by washing in PBS (6 times, 5 min). Avidin-biotin-peroxidase complex (ABC-Elite, Vector Laboratories) at a dilution of 1:100 was applied for 50 min. After washing in PBS (6 times, 5 min), the coloring reaction was carried out with DAB, and nuclei were counterstained with hematoxylin.

## In vitro erythroid differentiation

E13.5 fetal liver cells were harvested and maintained as described previously[91]. Briefly, lineage-negative cells were magnetically sorted using Lineage Cell Depletion Kit (Miltenyi Biotec). Then cells were seeded in fibronectin-coated plates (2 µg/cm$^2$, FUJIFILM Wako Pure Chemical) and cultured in IMDM supplemented with 15% FBS (Gibco), 200 µg/mL holo-transferrin (Sigma−Aldrich), 10 µg/mL human insulin (Sigma−Aldrich), 2 mM L-glutamine (Nacalai Tesque), 10$^{-4}$ M β-mercaptoethanol, 100 U/mL of penicillin, 100 µg/mL of streptomycin, 2 U/mL recombinant human erythropoietin (Epoetin Alfa, Kyowa Hakko Kirin), and 500 nM 4-OHT (Sigma−Aldrich). After 1–2 days of incubation, the cells were harvested and used for downstream analyses.

## Comet assay

Comet assay was performed using CometAssay reagents (Trevigen) according to the manufacturer's instructions. Briefly, cultured erythroblasts were mixed with the molten Comet LMAgarose, and plated onto the CometSlide. Slides were cooled at 4 °C for 30 min and then immersed in Lysis Solution (Trevigen) containing 10% DMSO (Nacalai Tesque) at 4 °C for 3 h. Then slides were immersed in Alkaline Unwinding Solution (200 mM NaOH and 1 mM EDTA, pH > 13) at 4 °C for 1 h. Next, alkaline electrophoresis was performed in a cooled Alkaline Electrophoresis Solution (200 mM NaOH, 1 mM EDTA, pH > 13) using cooled CometAssay Electrophoresis System II (Trevigen, 21 V, 30 min). Slides were washed twice in distilled water and once in 70% ethanol, air-dried, stained with SYBR Gold (Thermo Fisher Scientific) at room temperature for 30 min, and washed in distilled water. Slides were then air-dried completely and mounted with Prolong Diamond Antifade Mountant (Thermo Fisher Scientific). Images were acquired using the fluorescence microscope BZ-X810 (Keyence) and analyzed using CASPLab software (v1.2.2, https://casplab.com/)[92] with default settings.

## Nuclear and cytoplasmic extraction

Subcellular fractionation was performed using NE-PER Nuclear and Cytoplasmic Extraction Reagents (Thermo Fisher Scientific) according to the manufacturer's instructions. Then nuclear and cytoplasmic lysates were mixed with TRIzol LS (Thermo Fisher Scientific) to isolate RNA.

## Luciferase reporter assay

HEK293T or HeLa cells were transfected with siRNAs using Lipofectamine RNAiMAX (Invitrogen). After overnight incubation, cells were further transfected with indicated reporter vectors using PEI "Max" (Polysciences). Then after 24 h of incubation, cells were lysed, and luciferase activity was determined with the Dual-Luciferase Reporter Assay system (Promega) and GloMax-Multi Detection

System (Promega). The calculation methods of relative luciferase activity were described in each figure legend.

## RNA secondary structure prediction

The mfold program (v4.7) was utilized for RNA secondary structure predictions[93].

## Nucleofection of Cas9-ribonucleoproteins (RNPs)

Lineage-negative cells from E13.5 fetal livers were harvested as described above. Nucleofection of Cas9-RNPs was performed using 4D-Nucleofector Core and X Unit (Lonza) and P3 Primary Cell 4D-Nucleofector X Kit (Lonza) according to the method described previously with slight modification[94]. Briefly, one or two Alt-R CRISPR-Cas9 crRNA-XTs for each gene and Alt-R CRISPR-Cas9 tracrRNA (Integrated DNA Technologies) were mixed, heated at 95 °C for 5 min, and cooled to room temperature. The annealed crRNA/tracrRNA duplexes were mixed with TrueCut Cas9 Protein v2 (Thermo Fisher Scientific) and incubated at room temperature for 20 min. Then lineage-negative cells were mixed with Cas9-RNPs, transferred into the nucleofector cassette strip (Lonza), and immediately nucleofected (Program CM-138).

## Statistical analysis

Statistical analyses were performed using Prism v.8.2.1 (GraphPad) unless otherwise specified. Statistical significance was calculated with two-tailed Student's $t$-test. $P$-values of less than 0.05 were considered significant.

## Reporting summary

Further information on research design is available in the Nature Research Reporting Summary linked to this article.

## Data availability

The raw sequencing data generated in this study have been deposited in the DDBJ Sequence Read Archive under accession codes DRA013293, and DRA013294. The processed data reported in this paper are provided in the Supplementary Data files. All data supporting the findings of this study are present in the article and/or its Supplementary Information files. The *Mus musculus* mouse (GRCm38/mm10) reference genome and annotations used in this study are available from the GENCODE (https://www.gencodegenes.org/mouse/release_M10.html) or UCSC (https://hgdownload.soe.ucsc.edu/goldenPath/mm10/) servers. The processed RNA-seq data published by Kuppers et al.[57] are available in the NCBI Gene Expression Omnibus under accession number GSE106124. Source data are provided with this paper.

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

## Acknowledgements

We thank all members of our laboratory and Y. Ishigami (Cold Spring Harbor Laboratory) for helpful discussion and critical reading of the manuscript; Y. Okumoto for secretarial assistance; H. Miyachi, S. Kitano, K. Degawa, M. Idoji, T. Kubo, T. Kondo, Y. Sando, T. Tsujimura, T. Yamamoto, and S. Tarumoto (Kyoto University), K. Abe, T. Horiuchi, and K. Imamura (The University of Tokyo), Single-cell Genome Information Analysis Core (SignAC) at WPI-ASHBi, Kyoto University for technical assistance and RNA sequencing; U. Klingmuller (German Cancer Research Center), H. Kato, T. Fujita, K. Masuda, and H. Kawamoto (Kyoto University) for mice; O. Tanabe and M. Suzuki (Tohoku University) for mice and erythroid cell analyses; K. Terai (Kyoto University) for providing expertise for imaging; M. Kobayashi (Kyoto University) for cytospin slide preparations; the Center for Anatomical, Pathological and Forensic Medical Research, Kyoto University for preparing microscope slides. This research is supported by JSPS KAKENHI (18H05278, 16H06279 (PAGS), O.T.; 20K22737, 21K15372, M.Y.), AMED (#JP18am0101120 support number 0985, #21ae0121030h0001, O.T.), the joint research program of the Institute for Molecular and Cellular Regulation, Gunma University (20001, O.T., M.Y.), Fujiwara Memorial Foundation, Takeda Science Foundation, Shimizu Foundation for Immunology and Neuroscience, Biolegend and TOMY Digital Biology (M.Y.).

## Author contributions

M.Y. and O.T. conceived the project and designed the experiments. M.Y. performed most of the experiments. M.Y., M.C.B., and O.T. analyzed the data. S.Y., A.K., T.C., and Y.S. helped with experiments. K.H., D.W.M., and M.C.B. helped with the CRISPR screening experiments. P.H.C.C. and Y.K. provided expertise and helped with m6A analysis. M.Y., K.H., D.W.M., F.H., and A.V. performed the bioinformatics analysis. T.T. performed the histological and cytological analysis. T.H., R.K., and I.H. generated *Mettl16*-floxed mice. M.Y. and O.T. wrote the manuscript with input from all authors. O.T. supervised the project.

## Competing interests

The authors declare no competing interests.
