## [Peer Review File · Nature Communications]

The N6-methyladenosine methyltransferase METTL16 enables erythropoiesis through safeguarding genome integrityREVIEWER COMMENTS

Reviewer #1 (Remarks to the Author):

Epitranscriptomic regulation of gene expression mediated by RNA modification has been extensively studied in various biological contexts. In this manuscript, the authors conducted a genome-wide CRISPR screening for genes that modulate erythroid differentiation of K562 cells, and identified several genes that regulate TfR1 expression. Among them, they focused on METTL16 as a positive regulator for TfR1 expression. METTL16 is an evolutionarily conserved m6A writer for U6 snRNA and Mat2a mRNA. Compared to METTL3 writer complex, substrate specificity of METTL16 is limited, because METTL16 targets a specific nonamer sequence in structured RNA. The authors confirmed that the methyltransferase activity of METTL16 is required for TfR1 expression, indicating that METTL16-mediated m6A formation plays a critical role in TfR1 expression during erythroid differentiation. Curiously, steady-state level of Mettl16 mRNA is upregulated in the middle stage of erythroid differentiation. Next, the authors established erythroblast-specific Mettl16 knockout mouse (Mettl16^{fl/fl}Epor-Cre⁺) and found prenatal lethality of the knockout mice due to lower expression of TfR1 and impaired erythroid differentiation. Mettl16-deficient R3 erythroblasts not just showed altered gene expression patterns, but also displayed features of cells exposed to DNA damaging, such as emergence of micronuclei and expression of γ H2AX, finally leading to apoptosis. They performed MeRIP-seq of in vitro erythroblast culture cells, and identified downregulated mRNAs with hypomethylation in Mettl16 deficient cells. They are enriched in DNA repair and cell cycle process-related genes including Brca2 and Fancm. These mRNAs are also downregulated in R3 erythroblasts from Mettl16^{fl/fl}Epor-Cre⁺ mice. The authors showed that these mRNAs were not affected by SAM deficiency caused by loss of MAT2A expression. The m6A peaks on these genes were enriched with CAG motifs, and m6A modifications at the center of this motif were shown to be downregulated upon Mettl16 deficiency by the SELECT method. Pairwise CRISPRi screening identified nuclear exosome factors as genes which cancel out the effect of METTL16 depletion against TfR1 expression. Nuclear exosome factors such as DIS3 and Mtr4 were validated to show negative genetic interaction with METTL16. Overall, the authors show that genomic stability essential for erythroid differentiation is established through inhibition of nuclear mRNA decay of responsible genes by METTL16-mediated m6A modification.

The significance of the work is extremely high, because this manuscript clearly demonstrated epitranscriptomic regulation of gene expression mediated by METTL16 and physiological impact of mRNA m6A modification on erythroid differentiation. I believe this manuscript is meritorious for publication in Nature communications, assuming several issues outlined below can be addressed.

Major comments

Brca2 and Fancm mRNAs are stabilized by METTL16-mediated m6A deposition in the ACAGAR box during erythroid differentiation. If this system has a physiological impact on mammalian erythroid differentiation, the ACAGAR box and local secondary structures might be conserved in vertebrates or mammals. Some phylogenetic analyses are necessary to discuss this issue.

METTL16-mediated m6A modification is a positive regulator for Brca2 and Fancm mRNAs. But, the authors do not specify the function of m6A in gene expression throughout the paper.

The nuclear exosome complex was shown to be responsible for downregulation of METTL16-targeted genes (Fig. 6e-i), but the evidence for direct destabilization is still weak. The difference in mRNA decay speed upon METTL16 depletion should be measured, by methods such as Actinomycin treatment coupled with time-course RT-qPCR or SLAM seq.

In Supplementary Figs. 6bc, it was shown that SAM treatment is sufficient to downregulate Mat2a but does not restore the expression of DNA-repair-related genes. It was also shown that deficiency in the METTL3 complex does not alter the expression of DNA-repair related genes. From these data, the authors suggest that METTL16 controls the abundance of these mRNA independent of SAM concentration, and also that because METTL3 utilizes SAM for methylation, the downregulation is not because of impaired m6A deposition by the METTL3 complex. However, it has not been stated in the text nor cited from any paper that METTL3 complex activity is lowered upon SAM depletion, mentioning the kinetic parameters. In the last sentence of this paragraph, after explaining that METTL16 controls the expression of these genes independent of SAM concentration, it should be just stated that the mechanism of expression alteration is not mediated by m6A deposition to these mRNA by the METTL3 complex as an indirect effect of METTL16 depletion.

The authors claimed that MTR4-nuclear exosome complex is required for the METTL16-mediated mRNA regulation (Fig. 6e and f). However, the expression level of Tfr1 is drastically decreased upon MTR4 knockdown, resulting in the apparent decrease in the percent reduction of Tfr1 upon double knockdown with METTL16. Therefore, these data do not support the cooperative involvement of MTR4-nuclear exosome in METTL16-mediated mRNA regulation. Rather, this reviewer considers that MTR4 promotes the expression of Tfr1. How do the authors think about the alternative interpretation in Fig. 6e?

In discussion, the authors mentioned m1A58 in tRNA to explain stabilization effect of a single methylation on RNA structure. However, loss of m1A58 induces poly A tailing then recruit nuclear exosome complex. Thus, this instance is not appropriate here. Instead, it's better to mentioning a thermodynamic property of m6A, because syn conformation is slightly favored over the anti-conformation. Thus, m6A destabilizes A-U pair, but stabilizes A-A pair. In addition, m6A also has a function to stabilize RNA duplex, if m6A is present at the overhanging position. The chemical nature of m6A might be a clue to stabilize the target mRNAs. The authors should expand the discussion part with appropriate references.

Minor comments

- "N" of N6-methyladenosine should be italic.
- One gene should be stated in one name. Especially, it is confusing to have two or more names mixed in one figure like Fig. S1b.
- This reviewer does not understand the importance of the DFO treatment. Please explain this part more clearly.
- Fig. S1f. In the gene list in the legend, SLC11A2(DMT1) is missing.
- Fig. S1c,e. Define x and y axis.
- Fig. 2 e-g. The format of the legend is not consistent with other legends. And the legend for Fig.2f is missing.

- Fig. 3e. Y axis show percentage of MN+ or MN+γH2AX+ cells. What is the denominator of this percentage?
- Fig. 3f. Where should we focus on? The difference of the between control and METTL16 deficient mouse should be described more concretely.
- Fig. 4c. This figure does not include any gene names or groups at all, so I don't see the point of including it.
- Fig. 5d. The description of red-letter and blue-letter should be defined (even if we can assume red-color represents METTL16-target motif and blue letter represents m6A modification site).
- Fig. S6b They quantify the relative mRNA level of Gata2 but from the previous description, Gata1 should be examined. The same comment can be added on Fig. 6i.
- Fig. S7a Define x and y axis.
- On line123, revise the reference from Fig. 1e to the appropriate figure (Fig.1f or Supplementary Fig. 1e).
- Check the last sentences in the legend of Supplementary Fig. 2. (d) would be (e) and (e) would be (f).

Reviewer #2 (Remarks to the Author):

Yoshinaga et al. reported an essential role of METTL16 in erythropoiesis, which is discovered by a genome-wide CRISPR screening and have been further confirmed in *Mettl16* CKO mouse model. Mechanistically, *Mettl16* mediated m6A deposition on DNA-damage response-associated mRNAs, including *Brca2* and *Fancc*, and thereby upregulated their expression through a MTR4-mediated manner. Collectively, this manuscript is potentially interesting and within the scope of Nature Communications. There are some points will need to be addressed before the paper can be considered for publication.

1. It is important to clarify whether METTL16 maintained proper erythropoiesis depending on its m6A methyltransferase activity. The authors show wide-type but not catalytic dead METTL16 could regulate the expression of *TfR1* in Sup Figs 2f. I would suggest them to move this figure to the main figure. Furthermore, more experiments should be conducted to demonstrate that wide-type but not catalytic dead METTL16 could promote erythropoiesis and genome integrity.
2. At the beginning of this manuscript, METTL16 is selected for further study because it can significantly regulate the expression of *TfR1*, which is critical for erythropoiesis. However, the authors failed to demonstrate how does METTL16 regulate the expression of *TfR1*. Does METTL16 mediated m6A deposition and gene expression regulation of *GATA1* or other upstream regulators?
3. In Figure 4, the authors conducted MeRIP-seq in *Mettl16^{fl/fl} CreERT2⁺* mice, because the number of erythroblasts obtained from *Mettl16^{fl/fl} Epor-Cre⁺* embryos was insufficient. They should confirm that depletion of *Mettl16* in *Mettl16^{fl/fl} CreERT2⁺* mice could also impair erythropoiesis, as that in *Mettl16^{fl/fl} Epor-Cre⁺* mice.
4. The direct binding of METTL16 to *Brca2* and *Fancc* mRNAs should be examined.
5. How does METTL16-mediated m6A methylation on *Brca2* and *Fancc* mRNAs affect their expression through MTR4, which is not canonical m6A-associated RBPs? Furthermore,

does MTR4 indeed mediate nuclear transport and nuclear degradation of the target mRNAs?
6. Does METTL16 also play a role in adult erythropoiesis?
7. Some of the gene names are inconsistent in the main text and figures, which is confusing. For example, IRP2 is used in the main text while IREB2 in Figure1, TfR1 is used in the main text while CD71 in Figure2. Please keep consistency, or at least annotate the gene alias where it firstly shows.

Reviewer #3 (Remarks to the Author):

The authors performed a CRISPR-KO screen in K562 selecting to identify regulators of TfR1 (CD71) expression in K562 cells. Among the hits they identify and validate METTL16 as a key regulatory of TfR1 expression and erythroid differentiation. They further attempt to uncover mechanism through molecule characterization of K562 cells and mouse erythroid progenitors, including the use of floxed Mettl16 alleles. The results clearly establish a role for METTL16 in TfR1 expression and erythroid differentiation. Further, the notion that METTL16 regulates a portion of protein coding genes, including DNA repair genes is interesting and novel. However, there are two major weaknesses (and many minor ones) that need to be addressed. First is whether METTL16 roles in promoting "genome integrity" (i.e., m6A marking of DNA repair genes) are of sufficient strength/penetrance to cause the erythroid phenocopies in K562 and erythroid progenitors. As shown in Figure 3, DNA damage and apoptosis only represents a small portion of cells, whereas loss of TfR1 and Ki67 expression appears much more penetrant. This could mean that a DNA damage checkpoint – presumably p53 mediated – is at work arresting cells and triggering apoptosis. However, the authors fail to provide experiments to substantiate this – whereby partial inhibition of DNA damage proteins or p53 reverses the effects of or phenocopies METTL16 loss. There is also no confirmation of loss of protein expression of METTL16 m6A targets that I can see or some other assay for DNA repair activity. These confirmations are critical since METTL16 m6A regulation has previously been associated with non-protein coding RNA – which the authors do not pursue -- but which are lurking in the background.

The second major issue arises from confirmation studies of the secondary genetic screen to identify METTL16 downstream effectors. The authors identify the MTR4-nuclear exosome and claim that it "...plays a critical role in METTL16-mediated mRNA regulation in erythroblasts". This would be a very interesting result. However, this is not demonstrated. The results only show a similar effect, which appears to be an "epistatic" interaction or a phenocopy. Further molecular evidence is needed to support the notion that MTR4 has a direct – as opposed to indirect -- role in m6A regulation. I make a suggestion for an easy experiment below.

Detailed critique by figure:

Figure 1: The authors conducted a genome wide pooled CRISPR-KO screen in K562 selecting for low and high expression of TfR1 (CD71) by flow sorting. The screen hits were enriched for RNA binding and modification proteins based on GO analysis. The authors chose to follow up with this group of genes settling on METTL16.

◇Insufficient details have provided in the methods and main text to enable replication of the CRISPR screen.

From the text: We focused on an RNA N6-methyladenosine (m6A) transferase enzyme METTL16 for further functional analyses since METTL16 ablation greatly reduced TfR1 expression

(Fig. 1e). Figure 1e. shows IREB2 KO not METTL16.

Figure 2: The authors show elevated expression of METTL16 in MEP and erythroid populations compared to other hematopoietic progenitors and mature cell populations. They generate a flox KO mouse for METTL16 and use Epor driven Cre to generate erythroid specific KO mice. KO is embryonic lethal following E11.5. In KO mice, E10.5 erythroid progenitors maintain a larger cell size, suggesting a block to differentiation.

◇No information provided on the flow approach/surface markers used to define progenitor populations for Fig 2a.

Figure 3: The authors show that there was minimal transcriptional change in earlier erythroid progenitor populations following METTL16 KO. Significant down regulation of erythroid genes and up regulation of myeloid genes was observed following METTL16 KO in later erythroid progenitor populations. GSEA analysis revealed up regulation of DNA damage response genes in response to METTL16 KO. Increase cleaved caspase-3 staining, gH2AX and micronuclei were observed in the METTL16 KO cells. IFN signaling was activated in the METTL16 KO cells, likely in response to the micronuclei. Inhibition of IFN signaling did not rescue the erythroid phenotype.

- Rather than R2/R3, erythroid populations should be identified in the paper based on the flow criteria used to define them.
- It's not clear from the text what embryonic day the fetal liver cells were harvested from for all data presented in Fig. 3.
- It's not clear from the methods how the RNAseq data was analyzed.
- It's not clear what mRNA expression in Fig. 3b is relative to.
- More detail needs to be provided on how MN and gH2AX was quantified.

Figure 4: E13.5 fetal liver cells from Mettl16 CreERT2 mice were in vitro differentiated in the presence of EPO for two days in the presence or absence of 4-OHT and then MeRIP-seq performed. Among the genes with altered m6A marking, the majority were hypomethylated. The distribution of the sites with altered m6A marking on mRNA was similar to METTL3. The sites did not fit the DRACH motif, but rather were enriched for -CAG/-AG. DNA repair genes were enriched in gene hypomethylated following METTL16 KO. They demonstrate that the effect of METTL16 on erythropoiesis isn't due to METTL16 control of SAM levels via MAT2A.

• Based on the Epor mice, E11.5 or E12.5 erythroid progenitors would be the appropriate population for assessing the effect of Mettl16 KO on m6A marking. It's unclear what erythroid progenitor populations are represented by the E13.5 in vitro differentiated erythroid cells. No flow data or other assessment of the phenotypic effect of METTL16 KO during in vitro differentiation is provided.

• GO or GSEA analysis should be done on the three different groups presented in the venn diagram in Fig 4i in addition to calling out select DNA repair genes.

However, the SAM treatment failed to restore the expression of DNA-repair-related genes such as Brca1, Brca2 and Fancm under Mettl16 deficiency (Supplementary Fig. 6b). The DNA-repair-related genes were not affected in erythroid cells lacking METTL3 or WTAP (Supplementary Fig. 6c), confirming that DNA-damage-related genes are controlled by METTL16 in a mechanism distinct from the supply of SAM, which acts as a methyl donor required for general m6A modification by the METTL3 complex.

• The statement about METTL3/WTAP is using the HEL cell data. It's unclear if comparing a human cell line to mouse cells is a valid comparison, since it hasn't been established that the same mechanism is relevant during human erythropoiesis.

Figure 5: The authors demonstrate that there are Mettl16 regulated m6A sites in DNA repair genes and through Mettl16 KO/KD experiments that m6A marking regulates their expression.

- It's unclear exactly what cells are being used for these experiments. Presumably the E13.5 in vitro differentiated cells used for MeRIPseq. The same concerns about the use of these cells applies here, but otherwise the data supports the authors' conclusions.
- Use of IGV plots for Fig. 5a isn't appropriate since for a direct comparison the IP peaks should be normalized to input amounts.
- Except for Mat2a, the examples presented in Fig. 5d likely don't apply to human given sequence differences with mouse.

Figure 6: The authors performed a CRISPRi pairwise genetic interaction screen between METTL16 and the top 500 candidate regulators of CD71 from their initial CRISPR screen in K562 cells. They identify MTR4 as a top hit in the screen and aim to show that regulation of mRNA by METTL16 requires MTR4 and the MTR4-nuclear RNA exosome complex. The data shows that KO of a key component of the nuclear RNA exosome complex phenocopies METTL16 KO. However, it doesn't address the role of m6A in mediating this potential interaction.

- "confirmed that METTL16 knockdown failed to reduce TfR1 expression in MTR4-depleted cells (Figs. 6e, 6f)"
 - o CD71 levels following MTR4 KO are already reduced to the levels seen with METTL16 KO. Biologically it may not be possible to further reduce them with the addition of METTL16.
 - o Fig. 6f doesn't include METTL16.
- Fig. 6f,h: "Change in TfR1 expression levels in the presence and absence of METTL16 were summarized (f, h)" The data as presented only seems to show MTR4 or DIS3 KO in the presence of METTL16.
- Fig. 6i: "Mtr4 ablation reduced the effect of METTL16 ablation on the expression of METTL16 substrate mRNAs such as Mat2a, Fancm and Brca2, and also on erythroid-important genes including Klf1 and TfR1."
 - o The data shows no significant effect on expression of these genes, with METTL16 KO following Mtr4 KO, rather than a reduced effect.

While the results of the secondary functional genetic screen shown in Figure 6 are potentially interesting, the results suggest but fail to demonstrate that MTR4 plays roles in METTL16 function. Thus, the claim that "the MTR4-nuclear exosome plays a critical role in METTL16-mediated mRNA regulation in erythroblasts" is not demonstrated. The results only show a similar effect that appears to be an "epistatic" interaction or at least phenocopies but further molecular evidence is needed to support the notion that MTR4 has a direct – as opposed to indirect -- role in m6A regulation. One simple approach to further suggest this would be to perform luciferase reporter gene assays from Fig 5 in the context of MTR4 inhibition.

Supplementary Fig. 4: DNA damage under METTL16 deficiency impairs erythroid differentiation.

◇ There is no "DNA damage" in these experiments/panels.

A similar paper was published a few years ago on roles of m6A during erythropoiesis also using a CRISPR based approach. The authors fail to mention this work in the intro or discussion sections, yet integrate the data from the Koppers et al paper in their studies. The authors should highlight previous knowledge of m6A regulation of erythropoiesis.

We thank the reviewers for their constructive comments. Below, please find our point-by-point responses to all comments and questions. We believe that we have examined each of the points raised and could respond thoroughly. As suggested, we have added extensive data. Again, we would like to thank all the reviewers for being interested in our manuscript and for their thoughtful suggestions, which have definitely made this a stronger paper.

Reviewer #1 (Remarks to the Author):

The significance of the work is extremely high, because this manuscript clearly demonstrated epitranscriptomic regulation of gene expression mediated by METTL16 and physiological impact of mRNA m6A modification on erythroid differentiation. I believe this manuscript is meritorious for publication in Nature communications, assuming several issues outlined below can be addressed.

We thank the reviewer for considering our study highly important and for constructive comments. For the concerns raised by the reviewer, we responded to them as follows.

Major comments

1. Brca2 and Fancm mRNAs are stabilized by METTL16-mediated m6A deposition in the ACAGAR box during erythroid differentiation. If this system has a physiological impact on mammalian erythroid differentiation, the ACAGAR box and local secondary structures might be conserved in vertebrates or mammals. Some phylogenetic analyses are necessary to discuss this issue.

According to the reviewer's comment, we investigated the evolutionary conservation of the METTL16 target motifs among mammals. First, the ACAGAR boxes we identified in *Brca2* and *Fancm* mRNAs shown in Figure 5d are not widely conserved among mammalian species (**Figure R1** below).

Figure R1. Multiple alignments of *FANCM* and *BRCA2* mRNA sequences

FANCM and *BRCA2* mRNA sequences from mammals were aligned using the ClustalOmega algorithm (<https://www.ebi.ac.uk/Tools/msa/clustalo/>). The mouse ACAGAR motifs shown in Fig. 5d were highlighted in blue. Conserved sequences were shown in yellow. Dog, *Canis lupus familiaris*; hamster, *Mesocricetus auratus*; human, *Homo sapiens*; mouse, *Mus musculus*; rabbit, *Oryctolagus cuniculus*; rat, *Rattus norvegicus*.

Nevertheless, the regulation of DNA-damage-associated mRNAs by METTL16 appears to take place in human cells as well based on the following observations. First, we found that human *FANCM* and *BRCA2* mRNAs are downregulated in METTL16-deficient K562 cells (**Figure R2** below). Second, we found that several additional secondary structures that contain ACAGAR boxes are identified in human *BRCA2* and *FANCM* mRNAs (**Figure R3** below). Thus, we speculate that regulatory mechanisms for DNA damage-associated mRNAs via METTL16 may be conserved among mammals, although the METTL16 target motifs are not conserved. This is reminiscent of the evolutionary conservation of the regulation of SAM-producing enzymes (*MAT2A* in mammals and *sams* in worms) by METTL16. Although the METTL16 target sites and regulatory mechanisms are different between mammals and worms, METTL16 commonly regulates the abundance of mRNAs encoding SAM-producing enzymes in response to SAM concentrations (Pendleton *et al.*, Cell. 2017; Shima *et al.*, Cell Rep. 2017; Mendel *et al.*, Cell 2021; Watabe *et al.*, EMBO J. 2021). Since it has been shown that murine and human methylomes are substantially distinct (Liu *et al.*, Mol Cell. 2020), functional conservation

of m⁶A-mediated regulation, rather than sequences themselves, may be prevalent, which warrants further study. We added the following discussion on page 18 in the main text. “We found the structured motifs targeted by METTL16 on murine *Brca2* and *Fancm* mRNAs. However, it appears that these motifs are not conserved among species, especially in humans and mice. It is of note that the sequence conservation is not always seen in METTL16 targets as in the case of SAM-producing enzymes (*MAT2A* in mammals and *sams* in worms). Mammalian and worm METTL16 commonly regulate SAM-producing enzymes in response to SAM concentrations without the conservation of METTL16-target sites and regulatory mechanisms (Mendel *et al.*, 2021; Pendleton *et al.*, 2017; Shima *et al.*, 2017; Watabe *et al.*, 2021). Since it has been known that murine and human methylomes are substantially distinct (Liu *et al.*, 2020), functional conservation of m⁶A-mediated regulation, rather than sequences themselves, may be prevalent. It would be interesting to investigate METTL16 targets associated with DNA damage in human cells in future.”

Figure R2. *FANCM* and *BRCA2* mRNAs are downregulated in METTL16-deficient human K562 cells

K562-CRISPRi cells were transduced with lentiviral vectors encoding the indicated gRNAs. Then transduced cells were selected by puromycin, and total RNA was harvested and used for cDNA synthesis. mRNA levels were determined by RT-qPCR analysis and normalized to *ACTB* mRNA (n=3). ***p < 0.001 in Student’s t test. Data are expressed as mean ± SD.

Figure R3. Potential secondary structures containing ACAGAR boxes found in human *BRCA2* and *FANCM* mRNAs

mRNA secondary structures were predicted using mFold (Zuker, NAR 2003). Potential m⁶A methylated sites within ACAGAR boxes were highlighted by red circles.

2. *METTL16*-mediated m⁶A modification is a positive regulator for *Brca2* and *Fancm* mRNAs. But, the authors do not specify the function of m⁶A in gene expression throughout the paper. The nuclear exosome complex was shown to be responsible for downregulation of *METTL16*-targeted genes (Fig. 6e-i), but the evidence for direct destabilization is still weak. The difference in mRNA decay speed upon *METTL16* depletion should be measured, by methods such as Actinomycin treatment coupled with time-course RT-qPCR or SLAM

seq.

We thank the reviewer for the valuable suggestion to determine mRNA decay rates under METTL16 deficiency. However, we feel that it is quite difficult to assess the kinetics of the decay of mRNAs regulated by METTL16-mediated m⁶A deposition in the nucleus by using Actinomycin D treatment or SLAM-seq. Our data show that the MTR4-nuclear exosome is localized in the nucleus, and nuclear RNA processing and/or turnover via nuclear exosome is rapidly happening (Kilchert *et al.*, Nat Rev Mol Cell Biol. 2016; Ogami *et al.*, Noncoding RNA 2018). Further, processed mature mRNAs are immediately exported from the nucleus to the cytoplasm where the nuclear exosome components cannot access. Due to the dynamic nature of nuclear mRNA regulation, it is not possible to precisely quantify the decay of mRNAs in the nucleus by the kinetics analysis.

Transcribed pre-mRNAs undergo splicing and further processed mRNAs traffic to the cytoplasm. To investigate the step(s) of mRNA metabolism controlled by METTL16, we analyzed the expression of DNA damage-related genes in the nucleus and cytoplasm in METTL16-deficient cultured erythroblasts. The analysis revealed that nuclear pre-mRNA (unspliced) levels of *Brca2* and *Fancm* genes were comparable between METTL16-sufficient and -deficient cells, whereas spliced mRNAs of *Brca2* and *Fancm* genes in the nucleus and cytoplasm were significantly reduced in METTL16-deficient cells (**new Supplementary Fig. 13a**). These results strongly suggest that the regulation of DNA damage-associated mRNA mediated by METTL16 takes place in the nucleus following splicing, whereas their transcriptions are not affected by METTL16. The finding is consistent with the potential degradation of mRNAs via the MTR4-nuclear RNA exosome under METTL16 deficiency. We have added the new data and description on page 15 in the main text.

3. In Supplementary Figs. 6bc, it was shown that SAM treatment is sufficient to downregulate Mat2a but does not restore the expression of DNA-repair-related genes. It was also shown that deficiency in the METTL3 complex does not alter the expression of DNA-repair related genes. From these data, the authors suggest that METTL16 controls the abundance of these mRNA independent of SAM concentration, and also that because METTL3 utilizes SAM for methylation, the downregulation is not because of impaired m6A deposition by the METTL3 complex. However, it has not been stated in the text nor cited from any paper that METTL3 complex activity is lowered upon SAM depletion, mentioning the kinetic parameters. In the last sentence of this paragraph, after explaining

that METTL16 controls the expression of these genes independent of SAM concentration, it should be just stated that the mechanism of expression alteration is not mediated by m⁶A deposition to these mRNA by the METTL3 complex as an indirect effect of METTL16 depletion.

We thank the reviewer for this important comment. Following the reviewer's suggestion, we added the citation that showed the effect of SAM and MAT2A on global m⁶A methylation levels, although the kinetic parameters were not determined (Villa *et al.*, Mol Cell. 2021). Also, we have revised this paragraph as suggested by the reviewer on page 11 in the main text.

Moreover, to investigate the role of METTL3 in the regulation of DNA-damage-associated mRNAs, we deleted METTL3 in primary cultured erythroblasts by using the CRISPR/Cas9 system (**new Supplementary Fig. 10a**). We found that the depletion of METTL3 leads to only a modest reduction of mRNA levels of genes such as *Klf1*, *Fancm* and *Brca2*, unlike METTL16 deficiency (**new Supplementary Fig. 10b**). This finding strengthens the notion that the role of METTL16 in the regulation of erythropoiesis and DNA repair is distinct from that of the METTL3 complex. We added this new data and description on page 12 in the main text.

4. The authors claimed that MTR4-nuclear exosome complex is required for the METTL16-mediated mRNA regulation (Fig. 6e and f). However, the expression level of TfR1 is drastically decreased upon MTR4 knockdown, resulting in the apparent decrease in the percent reduction of TfR1 upon double knockdown with METTL16. Therefore, these data do not support the cooperative involvement of MTR4-nuclear exosome in METTL16-mediated mRNA regulation. Rather, this reviewer considers that MTR4 promotes the expression of TfR1. How do the authors think about the alternative interpretation in Fig. 6e?

We thank the reviewer for pointing out this issue. We share this concern with the reviewer and therefore, we performed several new experiments to investigate the potential involvement of MTR4-nuclear exosome in METTL16-mediated mRNA regulation. First, we conducted control experiments using IRP2, whose depletion downregulates TfR1 similarly to MTR4 (Effect size = -4.5 and -4.6, respectively, **Supplementary Data 1**). Noteworthy, we did not find a genetic interaction of IRP2 with METTL16 (T-score = 1.24, genes with GI score > 3 are considered to be strong genetic interactors, **Supplementary**

Data 6). We found that the depletion of IRP2 in METTL16-deficient cells further reduced TfR1 expression, unlike MTR4 depletion, in K562 cells (**Figure R4a** below). Moreover, the knockdown of METTL16 in IRP2 depleted cells reduced TfR1 expression in a similar manner to control cells (**Figure R4b** below). These results indicate that the expression of TfR1 was non-redundantly regulated by METTL16 and IRP2 whose genetic interactions are absent, whereas MTR4 knockdown failed to further suppress TfR1 expression in METTL16-depleted cells harboring genetic interactions (**Figures 6e-h**).

Figure R4. The effect of double depletion of METTL16 and IRP2 on TfR1 expression in K562 cells

K562-CRISPRi cells were transduced with the lentiviral vector expressing the indicated gRNA combinations. Then cells were treated with DFO overnight, stained with anti-TfR1 antibody and analyzed using flow cytometry (a). Change in TfR1 expression levels in the presence and absence of METTL16 were summarized (b). Each symbol represents the value from independent experiments. Horizontal lines indicate the mean. * $p < 0.05$ in Student's t test. ns, not significant.

Second, to investigate the involvement of the MTR4-nuclear exosome in METTL16-mediated mRNA regulation, we performed a new luciferase reporter assay. We depleted MTR4 or DIS3 together with METTL16 and checked the luciferase activity of the reporter harboring the METTL16-target sequence in the *Brca2* gene (used in Figure 5g). We found that the depletion of METTL16 failed to further suppress the *Brca2* reporter activity when MTR4 or DIS3 were depleted (**new Supplementary Fig. 13c**). This finding also supports the notion that the MTR4-nuclear exosome is the critical mediator that acts downstream of METTL16. We have added this new data and the corresponding description on page 15 in the main text.

Taken together, these findings suggest that MTR4 regulates the expression of

METTL16-target mRNAs in addition to its role in regulating TfR1 expression itself. It would be interesting to investigate the role of MTR4 itself in TfR1 expression, but we believe that this is beyond the scope of this study since this part is unrelated to the investigation of METTL16 and m⁶A functions.

5. In discussion, the authors mentioned m1A58 in tRNA to explain stabilization effect of a single methylation on RNA structure. However, loss of m1A58 induces poly A tailing then recruit nuclear exosome complex. Thus, this instance is not appropriate here. Instead, it's better to mentioning a thermodynamic property of m6A, because syn conformation is slightly favored over the anti-conformation. Thus, m6A destabilizes A-U paper, but stabilizes A-A pair. In addition, m6A also has a function to stabilize RNA duplex, if m6A is present at the overhanging position. The chemical nature of m6A might be a clue to stabilize the target mRNAs. The authors should expand the discussion part with appropriate references.

We thank the reviewer for the insightful comment. According to the reviewer's suggestion, we have revised the discussion section with references on page 17 in the main text.

Minor comments

- *“N” of N6-methyladenosine should be italic.*

We thank the reviewer for this comment. This error has been corrected throughout the manuscript in accordance with the reviewer's comment.

- *One gene should be stated in one name. Especially, it is confusing to have two or more names mixed in one figure like Fig. S1b.*

We thank the reviewer for this comment. We have revised the main text and figures using consistent gene names.

- *This reviewer does not understand the importance of the DFO treatment. Please explain*

this part more clearly.

According to the reviewer's comment, we have revised this part to clarify the role of DFO as follows in the Supplementary Figure legend.

“DFO treatment minimizes the effect of the change in intracellular iron concentration, which indirectly influences TfR1 expression.”

• *Fig. S1f. In the gene list in the legend, SLC11A2(DMT1) is missing.*

We thank the reviewer for this comment. This error has been corrected.

• *Fig. S1c,e. Define x and y axis.*

We have added the x (Enrichment) and y (Frequency) axis following the reviewer's suggestion.

• *Fig. 2 e-g. The format of the legend is not consistent with other legends. And the legend for Fig.2f is missing.*

We thank the reviewer for pointing out this issue. We have revised the figure legend for consistency with other legends. Also, we added the legend for Fig. 2f.

• *Fig. 3e. Y axis show percentage of MN⁺ or MN⁺ γ H2AX⁺ cells. What is the denominator of this percentage?*

The denominator of this percentage is all the cells we have analyzed. We added the corresponding description to the figure legend.

• *Fig. 3f. Where should we focus on? The difference of the between control and METTL16 deficient mouse should be described more concretely.*

We apologize for the poor presentation of these histological images. We have indicated cleaved caspase-3 positive cells with arrowheads (**new Figure 3f**).

• *Fig. 4c. This figure does not include any gene names or groups at all, so I don't see the point of including it.*

We thank the reviewer for this comment. Here, we found that the majority of the differentially methylated sites were hypomethylated under METTL16, which is consistent with the role of METTL16 as an m⁶A methyltransferase. Thus, we believe that this general tendency of methylation change under METTL16 deficiency is important information and should be shown. To deliver our interpretation more clearly, we grouped these genes into hypermethylated and hypomethylated and described in the figure.

• *Fig. 5d. The description of red-letter and blue-letter should be defined (even if we can assume red-color represents METTL16-target motif and blue letter represents m⁶A modification site).*

We thank the reviewer for pointing out this issue. We have clearly described the definition of colored letters in the figure legends.

• *Fig. S6b They quantify the relative mRNA level of Gata2 but from the previous description, Gata1 should be examined. The same comment can be added on Fig. 6i.*

We thank the reviewer for pointing out this issue. In these figures, we examined the *Gata2* mRNA expression as a negative control that is not affected by the depletion of METTL16 in erythroblasts. Nevertheless, we totally agree with the reviewer that *Gata1* mRNA expression is also important information to discuss the role of METTL16 in erythropoiesis. Following the reviewer's comment, we have examined the mRNA expression levels of *Gata1* in these conditions and added these figures in **new Supplementary Fig. 9b** and **Figure 6i**.

• *Fig. S7a Define x and y axis.*

We thank the reviewer for pointing out this error. The axes are now properly described in the **new Supplementary Figure 12a**.

• *On line 123, revise the reference from Fig. 1e to the appropriate figure (Fig. 1f or Supplementary Fig. 1e).*

According to the reviewer's suggestion, we have revised the reference in the main text.

• *Check the last sentences in the legend of Supplementary Fig. 2. (d) would be (e) and (e) would be (f).*

We thank the reviewer for the careful check of the manuscript. Now the figure has been moved to the main figure (**new Figures 1h and 1i**) according to the suggestion by the Reviewer #2 and the figure legend has been corrected as well.

Reviewer #2 (Remarks to the Author):

Yoshinaga et al. reported an essential role of METTL16 in erythropoiesis, which is discovered by a genome-wide CRISPR screening and have been further confirmed in Mettl16 CKO mouse model. Mechanistically, Mettl16 mediated m6A deposition on DNA-damage response-associated mRNAs, including Brca2 and Fancm, and thereby upregulated their expression through a MTR4-mediated manner. Collectively, this manuscript is potentially interesting and within the scope of Nature Communications. There are some points will need to be addressed before the paper can be considered for publication.

We thank the reviewer for considering our study interesting and for constructive comments.

1. It is important to clarify whether METTL16 maintained proper erythropoiesis depending on its m6A methyltransferase activity. The authors show wide-type but not

catalytic dead METTL16 could regulate the expression of TfR1 in Sup Figs 2f. I would suggest them to move this figure to the main figure. Furthermore, more experiments should be conducted to demonstrate that wide-type but not catalytic dead METTL16 could promote erythropoiesis and genome integrity.

We thank the reviewer for this constructive comment. According to the reviewer's suggestion, we moved the original Supplementary Fig. 2f to the main figure (**new Figures 1h and 1i**).

These data are consistent with a previous study showing that catalytic-dead METTL16 knock-in mice (PP185/186A or F187G) showed severe embryonic lethality as found in METTL16-deficient mice, supporting the importance of the catalytic activity of METTL16 in mouse development (Mendel *et al.*, Cell., 2021).

To further investigate if METTL16 has a methyltransferase activity-independent role in regulating gene expression following interaction with mRNAs, we tethered METTL16 to luciferase reporters without the ACAGAR boxes for methylation. We have shown that the expression of METTL16 potentiated the luciferase activity harboring the ACAGAR box from the *Brca2* and *Fancm* gene (**Figure 5g**). When we tethered wild-type or catalytic-dead METTL16 to a luciferase reporter without the ACAGAR box by utilizing the λ N-BoxB system (**Figure R5a** below), both wild-type and catalytic-dead METTL16 to 5' UTR or 3' UTR of the reporter failed to alter the luciferase reporter activity (**Figure R5b** below). Of note, the tethering of C terminus of TNRC6B (TNRC6B-Cterm) strongly suppressed the luciferase activity consistent with a previous study (Lazzaretti *et al.*, RNA 2009). These results suggest that the direct binding of METTL16 to target mRNAs is not sufficient to influence gene expression without the methylation of target mRNAs. Therefore, these results suggest that the catalytic activity of METTL16 is important for regulating gene expression.

Figure R5. The effect of the tethering of wild-type or catalytic-dead METTL16 to 5' UTR or 3' UTR.

A tethering assay was performed utilizing the λ N-BoxB system (Legault *et al.*, Cell 1998). HEK293T cells were transfected with the firefly luciferase reporter harboring either 3' UTR or 5' UTR 5x BoxB sequences (schematic representations are shown in **a**), together with the vectors expressing wild-type or catalytic-dead mutant METTL16 (PP185/186AA, F187G) fused with the λ N sequence. The C terminus of TNRC6B (TNRC6B-Cterm) was used as a positive control (Lazzaretti *et al.*, RNA 2009). Then cells were lysed, and luciferase activity was determined with the Dual-Luciferase Reporter Assay system (Promega) and GloMax-Multi Detection System (Promega). Renilla luciferase activity was used as an internal control. (**b**) Relative luciferase reporter activity tethered with indicated proteins (n=3 biological replicates). Data are expressed as mean \pm SD.

2. At the beginning of this manuscript, METTL16 is selected for further study because it can significantly regulate the expression of *TfR1*, which is critical for erythropoiesis. However, the authors failed to demonstrate how does METTL16 regulate the expression of *TfR1*. Does METTL16 mediated m6A deposition and gene expression regulation of *GATA1* or other upstream regulators?

We thank the reviewer for the important comments. Our MeRIP-seq data indicates that well-characterized regulators of erythroid differentiation, including GATA-1, KLF1 and TfR1 itself, are not differentially methylated under METTL16 deficiency (**Supplementary Data 4**). Moreover, according to this reviewer's comment, we performed MeRIP-qPCR analysis and found that GATA-1, KLF1 and TfR1 transcripts were not particularly enriched (**new Figure 5c**). These findings suggest that erythroid-important genes were not directly regulated by the methyltransferase activity of METTL16.

The indirect regulation of GATA-1 by METTL16 seems to play an essential role in the regulation of erythropoiesis, since GATA-1 is an essential transcription factor for the expression of erythroid-related genes such as TfR1 and KLF1 (Gammella *et al.*, Metallomics 2017; Crossley *et al.*, JBC 1994). In support of this notion, we found the downregulation of GATA-1 and upregulation of cleaved caspase-3 (**Figure 3f and Supplementary Figs. 5a and 5b**), which has been implicated in the degradation of GATA-1 in erythroblasts (De Maria *et al.*, Nature 1999; Ribeil *et al.*, Nature 2007; Tyrkalska *et al.*, Immunity 2019). Moreover, DNA damage-associated genes were greatly downregulated by the lack of METTL16, which can trigger the activation of caspase-3 (**Figure 5b and Supplementary Data 5**). Furthermore, the depletion of *Brca2* or *Fancc* gene in cultured erythroblasts reduced *Klf1* mRNA expression, a key transcription factor for erythroid gene expression (**new Supplementary Fig. 11c**). Therefore, we think that the failure in erythroid differentiation and gene expression such as TfR1 under METTL16 deficiency is initiated by the downregulation of DNA damage-associated genes and is mediated by the caspase-3-GATA-1 axis. We added this discussion on page 16 in the main text.

*3. In Figure 4, the authors conducted MeRIP-seq in *Mettl16fl/fl CreERT2+* mice, because the number of erythroblasts obtained from *Mettl16fl/fl Epor-Cre+* embryos was insufficient. They should confirm that depletion of *Mettl16* in *Mettl16fl/fl CreERT2+* mice could also impair erythropoiesis, as that in *Mettl16fl/fl Epor-Cre+* mice.*

We thank the reviewer for raising the point. According to the reviewer's suggestion, we investigated the cultured erythroblasts more extensively by performing flow cytometric analysis. We found that METTL16-deficient cultured erythrocytes showed impaired enucleation, which is a final event occurring in the erythropoiesis (**new Supplementary Figs. 7b and 7c**). This finding supports the notion that METTL16 is a critical regulator

of erythropoiesis in *Mettl16^{fl/fl}CreERT2⁺* mice.

4. *The direct binding of METTL16 to Brca2 and Fancm mRNAs should be examined.*

We thank the reviewer for this constructive comment. Following the reviewer's suggestion, we performed RNA immunoprecipitation (RIP) assay to verify the binding of METTL16 to *Brca2* and *Fancm* mRNAs. We found that wild-type METTL16 binds to *Brca2* and *Fancm* mRNA, to a similar extent to U6 snRNA, which is a well-characterized METTL16 substrate (**new Supplementary Fig. 11b**). The new data clearly demonstrate that *Brca2* and *Fancm* mRNA are directly targeted by METTL16.

5. *How does METTL16-mediated m⁶A methylation on Brca2 and Fancm mRNAs affect their expression through MTR4, which is not canonical m⁶A-associated RBPs? Furthermore, does MTR4 indeed mediate nuclear transport and nuclear degradation of the target mRNAs?*

We thank the reviewer for pointing out this issue. The reviewer asked how nuclear exosome discriminates m⁶A-modified and -unmodified mRNAs under METTL16-sufficient and -deficient conditions, respectively. We speculate that the chemical nature of m⁶A is likely to explain the underlying mechanisms. As the Reviewer #1 pointed out, m⁶A modification can alter secondary structures of mRNAs. We believe that such change can be detected by MTR4-nuclear exosome, thereby leading to the degradation of target mRNAs. Alternatively, additional m⁶A binding protein(s) (readers) may be required for the discrimination of modified and unmodified RNA, since it has been shown that MTR4 serves as an adaptor molecule for the nuclear RNA exosome and other RNA binding proteins that determine the target specificity, thereby mediating the degradation of target mRNAs (Zinder and Lima, *Genes Dev.* 2017; Ogami *et al.*, *Noncoding RNA* 2018). Although future studies are necessary to clarify the detailed mechanisms, our main goal of this study is to characterize the roles of METTL16 in erythropoiesis and its role in the regulation of DNA damage responses and we feel the MTR4-mediated regulatory mechanism is beyond the scope of this manuscript. Nevertheless, we discussed this point on page 17 in the main text.

The reviewer also asked about the fate of exosome-targeted mRNAs under

METTL16 deficiency. To investigate the step(s) of mRNA metabolism controlled by METTL16, we analyzed the expression of DNA damage-related genes in the nucleus and cytoplasm in METTL16-deficient cultured erythroblasts. The analysis revealed that nuclear pre-mRNA (unspliced) levels of *Brca2* and *Fancm* genes were comparable between METTL16-sufficient and -deficient cells, whereas spliced mRNAs of *Brca2* and *Fancm* genes in the nucleus and cytoplasm were significantly reduced in METTL16-deficient cells (**new Supplementary Fig. 13a**). These findings suggest that the degradation of these mRNAs takes place in the nucleus, following splicing. Since MTR4-nuclear exosome plays a critical role in the turnover of nuclear mRNA, it is conceivable that MTR4-nuclear exosome mediates nuclear degradation of METTL16-target mRNAs. We have added the new data and corresponding description on page 15 in the main text.

6. Does METTL16 also play a role in adult erythropoiesis?

We thank the reviewer for this important question. Since *METTL16^{fl/fl}Epor-Cre⁺* mice did not survive postnatally (**Figure 2c**), we took advantage of another mouse model crossed with Vav-Cre transgenic mouse (*METTL16^{fl/fl}Vav-Cre⁺* mice). We found that Vav-Cre activity failed to completely deplete METTL16 in hematopoietic cells during the embryonic stage as reported in a previous study using another floxed allele (Kerenyi *et al.*, eLife 2013). Consequently, they did not succumb to embryonic death unlike Epor-Cre, which allows us to investigate the role of METTL16 in erythropoiesis in the later developmental stage.

Nevertheless, *METTL16^{fl/fl}Vav-Cre⁺* mice died several days after birth. Hematological analysis of *METTL16^{fl/fl}Vav-Cre⁺* mice on the postnatal day 2-3 revealed that red blood cell number (RBC) and hemoglobin levels (HGB) were drastically decreased (**Figure R6** below), suggesting that *METTL16^{fl/fl}Vav-Cre⁺* mice suffer from severe anemia. Interestingly, the anemia was macrocytic (MCV is higher than controls), consistent with the notion that this anemia is associated with the failure of DNA synthesis and/or repair. Definitive erythropoiesis at this stage is considered to be adult-type and shares most of the aspects of adult erythropoiesis, which occurs in the adult bone marrow. Thus, we believe that these findings further highlight the importance of METTL16 in erythropoiesis throughout life.

Figure R6. METTL16 deficiency in hematopoietic cells leads to severe anemia postnatally

Blood samples were harvested from control and *METTL16^{fl/fl}Vav-Cre⁺* mice 2-3 days after birth. Hematological parameters were analyzed as described previously (Yoshinaga *et al.*, Cell Rep. 2017). RBC, red blood cell; HGB, hemoglobin; HCT, hematocrit; MCV, mean corpuscular volume; MCHC, mean corpuscular hemoglobin concentration. Each symbol represents the value from individual mice. Horizontal lines indicate the mean. ** $p < 0.01$, **** $p < 0.0001$ in Student's t test. ns, not significant.

7. Some of the gene names are inconsistent in the main text and figures, which is confusing. For example, *IRP2* is used in the main text while *IREB2* in Figure1, *TfR1* is used in the main text while *CD71* in Figure2. Please keep consistency, or at least annotate the gene alias where it firstly shows.

We thank the reviewer for pointing out this issue. We have thoroughly revised the main text and figures following the reviewer's suggestion. For transferrin receptor, since *CD71* is more commonly used for flow cytometric analysis, we would like to retain *CD71* and annotate it in the main text and Fig. 2e.

Reviewer #3 (Remarks to the Author):

The authors performed a CRISPR-KO screen in K562 selecting to identify regulators of TfR1 (CD71) expression in K562 cells. Among the hits they identify and validate METTL16 as a key regulatory of TfR1 expression and erythroid differentiation. They further attempt to uncover mechanism through molecule characterization of K562 cells and mouse erythroid progenitors, including the use of floxed Mettl16 alleles. The results clearly establish a role for METTL16 in TfR1 expression and erythroid differentiation. Further, the notion that METTL16 regulates a portion of protein coding genes, including DNA repair genes is interesting and novel.

We thank the reviewer for thoroughly evaluating our manuscript and for considering it interesting and novel.

However, there are two major weaknesses (and many minor ones) that need to be addressed. First is whether METTL16 roles in promoting "genome integrity" (i.e., m6A marking of DNA repair genes) are of sufficient strength/penetrance to cause the erythroid phenocopies in K562 and erythroid progenitors. As shown in Figure 3, DNA damage and apoptosis only represents a small portion of cells, whereas loss of TfR1 and Ki67 expression appears much more penetrant. This could mean that a DNA damage checkpoint – presumably p53 mediated – is at work arresting cells and triggering apoptosis. However, the authors fail to provide experiments to substantiate this – whereby partial inhibition of DNA damage proteins or p53 reverses the effects of or phenocopies METTL16 loss. There is also no confirmation of loss of protein expression of METTL16 m6A targets that I can see or some other assay for DNA repair activity. These confirmations are critical since METTL16 m6A regulation has previously been associated with non-protein coding RNA – which the authors do not pursue -- but which are lurking in the background.

The first major question is if the DNA damage is prevalently happening or not. As the reviewer pointed out, the frequency of micronuclei formation is approximately 10% in METTL16-deficient erythroblasts (**Figure 3d**). This may be because the DNA damage is not always detectable by the formation of micronuclei. Although micronuclei formation is suggestive of the emergence of the DNA damaging events such as the double-stranded break, but DNA damaging events do not always induce micronuclei formation. Indeed, a previous study has shown that the treatment of bone marrow erythroblasts with DNA damaging agents (BCNU, MNNG and MMS) led to micronuclei formation in

approximately 10% of all analyzed cells, although the treatment with DNA damaging agents reduced the viable cell numbers by 30-40% (Shuga *et al.*, PNAS 2007). Thus, it is plausible that the frequency of micronuclei formation is basically fewer than the that of DNA damage. In addition, formed micronuclei can overlap with the nucleus because of the high nuclear/cytoplasmic ratio of erythroblasts, which makes micronuclei detection by microscopy difficult. Therefore, we believe that the micronuclei found in METTL16-deficient erythroblasts resulted from widely prevalent DNA damage.

To further investigate the levels of DNA damage in METTL16-deficient erythroblasts, we conducted the alkaline comet assay, which is a test for detecting DNA damage and/or DNA repair activity more sensitively (Olive and Banath, Nat Protocols. 2006). In this assay, the DNA content and length of the “comet tail” represent the extent of single-stranded and/or double-stranded DNA breaks. We found that comets generated from METTL16-deficient primary cultured erythroblasts are significantly longer and contained more DNA than controls (**new Supplementary Figs. 8a, 8b**), indicating the increased amount of DNA lesions under METTL16 deficiency. These results suggest that DNA repair activity is substantially reduced in a large fraction of METTL16-deficient erythroblasts.

The reviewer also asked us if the loss of DNA damage-related proteins phenocopies METTL16 deficiency in erythroblasts. To address this question, we depleted *Brca2* or *Fancm* genes in the cultured erythroblasts by taking advantage of the nucleofection system (Lonza). We found that the ablation of these DNA damage-related proteins resulted in the downregulation of *Klf1* mRNA, a key transcription factor for the erythroid gene expression (**new Supplementary Fig. 11b**), which is consistent with the defect in erythroid differentiation observed under METTL16 deficiency (**Supplementary Fig. 9a**). Therefore, METTL16 deficiency impaired the erythroid differentiation possibly through the downregulation of DNA damage-associated genes. We think that the cumulative effect of the downregulation of many DNA-repair-related genes triggers more substantial impairment of erythroid gene expression as seen in METTL16 deficiency.

Moreover, the reviewer suggested us to check the expression of DNA-damage-associated genes under METTL16 deficiency at the protein levels. According to the reviewer’s suggestion, we performed the immunoblot analysis of BRCA2 and FANCM proteins. We found that these proteins were reduced in *Mettl16*-deficient cultured erythroblasts (**Supplementary Fig. 11a**), consistent with the reduced expression of *Brca2* and *Fancm* mRNAs under METTL16 deficiency.

With these new data, we would like to propose that erythroid defects found under METTL16 deficiency are initiated by widely prevalent DNA damage due to the

downregulation of DNA-repair-related genes.

The second major issue arises from confirmation studies of the secondary genetic screen to identify METTL16 downstream effectors. The authors identify the MTR4-nuclear exosome and claim that it "...plays a critical role in METTL16-mediated mRNA regulation in erythroblasts". This would be a very interesting result. However, this is not demonstrated. The results only show a similar effect, which appears to be an "epistatic" interaction or a phenocopy. Further molecular evidence is needed to support the notion that MTR4 has a direct – as opposed to indirect -- role in m6A regulation. I make a suggestion for an easy experiment below.

The second major concern raised by this reviewer is that we need more evidence to claim that the MTR4-nuclear exosome plays a role in METTL16-mediated mRNA regulation. According to the reviewer's suggestion, we depleted MTR4 or DIS3 together with METTL16 and investigated the luciferase activity of the reporter harboring the METTL16-target sequence in the *Brca2* gene (used in Figure 5g). We found that the depletion of METTL16 failed to further suppress the *Brca2* reporter activity when MTR4 or DIS3 were depleted (**new Supplementary Fig. 13**). This finding further supports the notion that METTL16 requires MTR4-nuclear RNA exosome for the regulation of target mRNA expression. We have added this new data and the corresponding description on page 15 in the main text.

Detailed critique by figure:

Figure 1: The authors conducted a genome wide pooled CRISPR-KO screen in K562 selecting for low and high expression of TfR1 (CD71) by flow sorting. The screen hits were enriched for RNA binding and modification proteins based on GO analysis. The authors chose to follow up with this group of genes settling on METTL16.

Insufficient details have provided in the methods and main text to enable replication of the CRISPR screen.

We thank the reviewer for this comment. For the CRISPR screen, we followed the experimental procedures as described previously (Deans *et al.*, Nat Chem Biol. 2016). Since the details of lentivirus transduction, sgRNA library preparation and data analysis were described in this reference, we have cited this article in the Methods section.

Moreover, we have revised the methods section for providing sufficient details in the methods for cell preparation and sorting, since we modified the methods from those described in the above-mentioned paper (Deans *et al.*, Nat Chem Biol. 2016).

From the text: We focused on an RNA N6-methyladenosine (m6A) transferase enzyme METTL16 for further functional analyses since METTL16 ablation greatly reduced TfR1 expression

(Fig. 1e). Figure 1e. shows IREB2 KO not METTL16.

We thank the reviewer for pointing out this error. We have corrected the figures using the data from METTL16 KO cells.

Figure 2: The authors show elevated expression of METTL16 in MEP and erythroid populations compared to other hematopoietic progenitors and mature cell populations. They generate a flox KO mouse for METTL16 and use Epor driven Cre to generate erythroid specific KO mice. KO is embryonic lethal following E11.5. In KO mice, E10.5 erythroid progenitors maintain a larger cell size, suggesting a block to differentiation. No information provided on the flow approach/surface markers used to define progenitor populations for Fig 2a.

We thank the reviewer for this comment. We have now added the representative FACS plot showing how the hematopoietic stem/progenitor cells were gated in **new Supplementary Fig. 3**.

Figure 3: The authors show that there was minimal transcriptional change in earlier erythroid progenitor populations following METTL16 KO. Significant down regulation of erythroid genes and up regulation of myeloid genes was observed following METTL16 KO in later erythroid progenitor populations. GSEA analysis revealed up regulation of DNA damage response genes in response to METTL16 KO. Increase cleaved caspase-3 staining, gH2AX and micronuclei were observed in the METTL16 KO cells. IFN signaling was activated in the METTL16 KO cells, likely in response to the micronuclei. Inhibition of IFN signaling did not rescue the erythroid phenotype.

· Rather than R2/R3, erythroid populations should be identified in the paper based on

the flow criteria used to define them.

We thank the reviewer for this comment. Following the reviewer's suggestion, we revised the main text based on the flow criteria (CD71⁺Ter119⁻ or CD71⁺Ter119⁺).

- *It's not clear from the text what embryonic day the fetal liver cells were harvested from for all data presented in Fig. 3.*
- *It's not clear from the methods how the RNAseq data was analyzed.*
- *It's not clear what mRNA expression in Fig. 3b is relative to.*
- *More detail needs to be provided on how MN and gH2AX was quantified.*

We thank the reviewer for these comments. We have revised these points following the reviewer's comments.

Figure 4: E13.5 fetal liver cells from Mettl16 CreERT2 mice were in vitro differentiated in the presence of EPO for two days in the presence or absence of 4-OHT and then MeRIP-seq performed. Among the genes with altered m6A marking, the majority were hypomethylated. The distribution of the sites with altered m6A marking on mRNA was similar to METTL3. The sites did not fit the DRACH motif, but rather were enriched for -CAG/-AG. DNA repair genes were enriched in gene hypomethylated following METTL16 KO. They demonstrate that the effect of METTL16 on erythropoiesis isn't due to METTL16 control of SAM levels via MAT2A.

- *Based on the Epor mice, E11.5 or E12.5 erythroid progenitors would be the appropriate population for assessing the effect of Mettl16 KO on m6A marking. It's unclear what erythroid progenitor populations are represented by the E13.5 in vitro differentiated erythroid cells. No flow data or other assessment of the phenotypic effect of METTL16 KO during in vitro differentiation is provided.*

We thank the reviewer for this comment. We have performed the flow cytometric analysis of the *in vitro* differentiated erythroblasts following the reviewer's comment. We found that these cells were CD71⁺Ter119⁺ cells as shown in the **new Supplementary Fig. 7a**, suggesting that these cells are corresponding to the CD71⁺Ter119⁺ erythroid cells in the fetal liver of E11.5 or E12.5 embryos.

As we have described in the main text, the number of CD71⁺Ter119⁺ erythroid

cells that can be isolated from the fetal livers of E11.5 embryos was very few (several thousand cells per embryo). For E12.5 embryos, less than half of the embryos survived at this stage (**Figure 2c**), and the number of CD71⁺Ter119⁺ erythroid cells from live embryos was similarly quite small. Therefore, it was technically challenging to obtain reliable data by using these cells. For this reason, we believe that *in vitro* differentiated erythroid cells are a more suitable model for analyzing the m⁶A status.

· *GO or GSEA analysis should be done on the three different groups presented in the venn diagram in Fig 4i in addition to calling out select DNA repair genes.*

According to the reviewer's suggestion, we performed GO analysis (for the molecular function GOs) of three different groups shown in Fig. 4i. This analysis revealed that DNA repair-related genes were greatly enriched in the intersect (**new Figure 4i**), and we were not able to find any significantly enriched GO molecular function terms in other two groups (right and left). These findings further support the notion that METTL16 regulates DNA damage response. We added the new data and corresponding description in the main text.

However, the SAM treatment failed to restore the expression of DNA-repair-related genes such as Brca1, Brca2 and Fancm under Mettl16 deficiency (Supplementary Fig. 6b). The DNA-repair-related genes were not affected in erythroid cells lacking METTL3 or WTAP (Supplementary Fig. 6c), confirming that DNA-damage-related genes are controlled by METTL16 in a mechanism distinct from the supply of SAM, which acts as a methyl donor required for general m⁶A modification by the METTL3 complex.

· *The statement about METTL3/WTAP is using the HEL cell data. It's unclear if comparing a human cell line to mouse cells is a valid comparison, since it hasn't been established that the same mechanism is relevant during human erythropoiesis.*

We thank the reviewer for pointing out this issue. To investigate the role of METTL3 in murine erythroblasts, we ablated METTL3 in the cultured fetal liver erythroblasts utilizing the Cas9-gRNA RNP nucleofection (Lonza) (**new Supplementary Fig. 10**). We found that the deletion of METTL3 in this setting only modestly affected the expression levels of *Mat2a*, *Brca2* and *Fancm* (**new Supplementary Fig. 10**), which is consistent with human HEL cell data. These new results further strengthen that METTL16 has a

nonredundant role in the expression of these genes, which is not overlapping with METTL3.

Figure 5: The authors demonstrate that there are Mettl16 regulated m6A sites in DNA repair genes and through Mettl16 KO/KD experiments that m6A marking regulates their expression.

· It's unclear exactly what cells are being used for these experiments. Presumably the E13.5 in vitro differentiated cells used for MeRIPseq. The same concerns about the used of these cells applies here, but otherwise the data supports the authors conclusions.

We apologize for the confusion caused by the insufficient experimental description. In the experiments shown in Figure 5, the E13.5 *in vitro* differentiated erythroid cells were consistently used. We have revised the figure legends accordingly. As we have described above, these cultured erythroblasts recapitulate the characteristic of the *in vivo* CD71⁺Ter119⁺ erythroblasts, and therefore we believe that they are a good substitute for analyzing the m⁶A landscape.

· Use of IGV plots for Fig. 5a isn't appropriate since for a direct comparison the IP peaks should be normalized to input amounts.

We thank the reviewer for this comment. However, we would like to show the m⁶A peaks and RNA-seq tracks at the same time, since this way of data presentation enables us to visualize the downregulation of both m⁶A and mRNA expression levels under METTL16 deficiency. This style of data presentation is often seen in many other papers related to RNA m⁶A methylation (Engel *et al.*, Neuron 2018; Tong *et al.*, Genome Biol. 2018). Furthermore, this finding is supported by the following qPCR results (**Figures 5b and 5c**).

· Except for Mat2a, the examples presented in Fig. 5d likely don't apply to human given sequence differences with mouse.

We thank the reviewer for pointing out this issue. The response to this comment is the same as the first comment from Reviewer #1, and so please find the response above

(pages 1-4 of this letter).

Figure 6: The authors performed a CRISPRi pairwise genetic interaction screen between METTL16 and the top 500 candidate regulators of CD71 from their initial CRISPR screen in K562 cells. They identify MTR4 as a top hit in the screen and aim to show that regulation of mRNA by METTL16 requires MTR4 and the MTR4-nuclear RNA exosome complex.

The data shows that KO of a key component of the nuclear RNA exosome complex phenocopies METTL16 KO. However, it doesn't address the role of m6A in mediating this potential interaction.

· “confirmed that METTL16 knockdown failed to reduce TfR1 expression in MTR4-depleted cells (Figs. 6e, 6f)”

o CD71 levels following MTR4 KO are already reduced to the levels seen with METTL16 KO. Biologically it may not be possible to further reduced them with the addition of METTL16.

We thank the reviewer for this comment. Because the comment 4) of Reviewer #1 overlaps with this comment, please find our detailed responses in that section (pages 6-8 of this letter).

o Fig. 6f doesn't include METTL16.

· Fig. 6f,h: “Change in TfR1 expression levels in the presence and absence of METTL16 were summarized (f, h)” The data as presented only seems to show MTR4 or DIS3 KO in the presence of METTL16.

We apologize for the confusion caused by the original figure legends. Here the change in TfR1 expression in the presence or absence of METTL16 is calculated. To make this point clear, we have revised the figure legends by showing how we calculated these values in detail.

· Fig. 6i: “Mtr4 ablation reduced the effect of METTL16 ablation on the expression of METTL16 substrate mRNAs such as Mat2a, Fancm and Brca2, and also on erythroid-important genes including Klf1 and TfR1.”

o The data shows no significant effect on expression of these genes, with METTL16 KO following Mtr4 KO, rather than a reduced effect.

We thank the reviewer for pointing out this issue. We have revised the main text following the reviewer's comment.

While the results of the secondary functional genetic screen shown in Figure 6 are potentially interesting, the results suggest but fail to demonstrate that MTR4 plays roles in METTL16 function. Thus, the claim that "the MTR4-nuclear exosome plays a critical role in METTL16-mediated mRNA regulation in erythroblasts" is not demonstrated. The results only show a similar effect that appears to be an "epistatic" interaction or at least phenocopies but further molecular evidence is needed to support the notion that MTR4 has a direct – as opposed to indirect -- role in m6A regulation. One simple approach to further suggest this would be to perform luciferase reporter gene assays from Fig 5 in the context of MTR4 inhibition.

We thank the reviewer for this constructive comment. Following the reviewer's suggestion, we performed the luciferase reporter assay using the *Brca2* mRNA reporter under the knockdown of MTR4 or DIS3, together with METTL16. As expected, the reporter expression was not significantly affected by the METTL16 knockdown if MTR4 or DIS3 knockdown was simultaneously performed (new **Supplementary Fig. 13c**). This new finding further supports the notion that the MTR4-nuclear exosome is involved in the METTL16-mediated mRNA regulation.

Supplementary Fig. 4: DNA damage under METTL16 deficiency impairs erythroid differentiation.

There is no "DNA damage" in these experiments/panels.

We thank the reviewer for pointing out this issue. We have revised the title of this figure.

A similar paper was published a few years ago on roles of m6A during erythropoiesis also using a CRISPR based approach. The authors fail to mention this work in the intro or discussion sections, yet integrate the data from the Koppers et al paper in their studies.

The authors should highlight previous knowledge of m6A regulation of erythropoiesis.

We apologize this citation has been dropped off during the manuscript editing process. Following the reviewer's comment, we have cited this article properly in the discussion section on page 16 in the main text.

REVIEWER COMMENTS

Reviewer #1 (Remarks to the Author):

The revised manuscript adequately addresses most of my concerns and comments. I have no further comments. Many congrats!

Reviewer #2 (Remarks to the Author):

The authors addressed most of my questions except for the major point 1. I persist it is important to demonstrate that Mettl16 promotes erythropoiesis and genome integrity depending on its methyltransferase activity. It should be noted that the embryonic lethality of METTL16 KI mice reported by other group could not demonstrate its function on erythropoiesis. The authors could have performed rescue experiments with wild-type and catalytic dead METTL16.

Reviewer #3 (Remarks to the Author):

The authors have sufficiently addressed one of the two major weaknesses I felt were apparent in the previous submission of this manuscript. The first was whether METTL16 roles in promoting "genome integrity" (i.e., m6A marking of DNA repair genes) are of sufficient strength/penetrance to cause the erythroid phenocopies in K562 and erythroid progenitors.

The addition of the Comet assay and also the western blots for down regulation of DNA repair genes significantly allay my concerns.

The second major issue arose from confirmation studies of from the secondary genetic screen to identify METTL16 downstream effectors. The authors identify the MTR4-nuclear exosome and claim that it "...plays a critical role in METTL16-mediated mRNA regulation in erythroblasts". This would be a very interesting result. However, this is not demonstrated. The results only show a similar effect, which appears to be an "epistatic" interaction or phenocopies. Further molecular evidence is needed to support the notion that MTR4 has a direct – as opposed to indirect -- role in m6A regulation.

The authors provide an additional experiment in HeLa cells to address whether a Brca2 reporter (I assume the same one in Fig 5f). The authors assert that "We found that the depletion of METTL16 failed to further suppress the Brca2 reporter activity when MTR4 or DIS3 were depleted (new Supplementary Fig. 13). This finding further supports the notion that METTL16 requires MTR4-nuclear RNA exosome for the regulation of target mRNA expression." However, one key to this experiment was using mutant forms of the reporter shown in figure 5g. The mutant forms would establish some basis to argue that MTR4 and DIS3 act through these sequences. Unfortunately, the authors do not provide new data to support the assertion that "METTL16 regulates the substrate mRNA expression in a manner dependent on the MTR4-nuclear RNA exosome complex in erythroblasts." Instead, the data support the notion the MTR4 complex has a role in regulating METTL16 target genes, but not that MTR4 activity is causal for METTL16 effects.

We thank the reviewers for their constructive comments. Below, please find our point-by-point responses to all comments and questions. We believe that we have examined each of the points raised and could respond thoroughly. As suggested, we have added extensive data. Again, we would like to thank all the reviewers for being interested in our manuscript and for their thoughtful suggestions, which have definitely made this a stronger paper.

Reviewer #1 (Remarks to the Author):

The revised manuscript adequately addresses most of my concerns and comments. I have no further comments. Many congrats!

We thank the reviewer for carefully evaluation of our responses, and for helpful suggestions to improve our manuscript and encouraging remarks.

Reviewer #2 (Remarks to the Author):

The authors addressed most of my questions except for the major point 1. I persist it is important to demonstrate that Mettl16 promotes erythropoiesis and genome integrity depending on its methyltransferase activity. It should be note that the embryonic lethality of METTL16 KI mice reported by other group could not demonstrate its function on erythropoiesis. The authors could performed rescue experiments with wide-type and catalytic dead METTL16.

We thank the reviewer for acknowledging that our revisions have mostly satisfactorily addressed the comments.

The reviewer asked about the role of the methyltransferase activity of METTL16 in our experimental models. To address this question, we took advantage of the human K562 cell line, which was utilized for our initial CRISPR screening experiments, to reconstitute wild-type and catalytic dead METTL16. K562 cells can produce fetal hemoglobin (HBG) upon hemin stimulation, which is considered to model the erythroid differentiation (Andersson, et al., Nature 1979). In METTL16-deficient K562 cells, the mRNA expression levels of HBG were decreased, suggesting that METTL16 is also important for human erythropoiesis (new Supplementary Fig. 2e). We transduced the lentiviruses expressing the gRNA-resistant wild-type or catalytic-dead mutant METTL16 into METTL16-deficient K562 cells, harvested the transduced cells by cell sorting, and stimulated with hemin for 24 h. We found that wild-type METTL16, but not catalytic-

dead mutants, restored the *HBG* mRNA expression levels (new Supplementary Fig. 2f). Together with Fig. 1i and figures shown in the previous response letter, this new finding further corroborates the notion that the catalytic activity of METTL16 is important for the regulation of erythroid differentiation. We have added these data and the corresponding description in the main text.

Reviewer #3 (Remarks to the Author):

The authors have sufficiently addressed one of the two major weaknesses I felt were apparent in the previous submission of this manuscript. The first was whether METTL16 roles in promoting "genome integrity" (i.e., m6A marking of DNA repair genes) are of sufficient strength/penetrance to cause the erythroid phenocopies in K562 and erythroid progenitors.

The addition of the Comet assay and also the western blots for down regulation of DNA repair genes significantly allay my concerns.

We thank the reviewer for carefully evaluating our revision and acknowledged our new data for showing the involvement of METTL16 in the control of genome integrity.

The second major issue arose from confirmation studies of from the secondary genetic screen to identify METTL16 downstream effectors. The authors identify the MTR4-nuclear exosome and claim that it "...plays a critical role in METTL16-mediated mRNA regulation in erythroblasts". This would be a very interesting result. However, this is not demonstrated. The results only show a similar effect, which appears to be an "epistatic" interaction or phenocopies. Further molecular evidence is needed to support the notion that MTR4 has a direct – as opposed to indirect -- role in m6A regulation.

The authors provide an additional experiment in HeLa cells to address whether a Brca2 reporter (I assume the same one in Fig 5f). The authors assert that "We found that the depletion of METTL16 failed to further suppress the Brca2 reporter activity when MTR4 or DIS3 were depleted (new Supplementary Fig. 13). This finding further supports the notion that METTL16 requires MTR4-nuclear RNA exosome for the regulation of target mRNA expression." However, one key to this experiment was using mutant forms of the reporter shown in figure 5g. The mutant forms would establish some basis to argue that MTR4 and DIS3 act through these sequences. Unfortunately, the authors do not provide new data to support the assertion that "METTL16 regulates the substrate mRNA

expression in a manner dependent on the MTR4-nuclear RNA exosome complex in erythroblasts.” Instead, the data support the notion the MTR4 complex has a role in regulating METTL16 target genes, but not that MTR4 activity is causal for METTL16 effects.

We thank the reviewer for reevaluating the manuscript and for the thoughtful comments. According to the reviewer’s suggestion, we performed the experiment shown in Supplementary Fig. 13c by using the wild-type and mutant *Brca2* reporter that was originally shown in Fig. 5f. We found that the wild-type *Brca2* reporter activity was decreased by METTL16 knockdown in an MTR4-dependent manner, when compared to mutant *Brca2* reporter (new Supplementary Fig. 13c). This new finding supports the notion that the MTR4 activity is causal for METTL16 effects. Nevertheless, we think that further investigation is necessary to clarify the mechanisms of how MTR4-nuclear exosome is involved in the METTL16-mediated mRNA regulation. Therefore, we revised the conclusion sentences in the abstract and main text on pages 2, 15, 16 and 17 accordingly.

REVIEWERS' COMMENTS

Reviewer #2 (Remarks to the Author):

The revised manuscript addresses most of my concerns. I have no further comments.

Again, we would like to thank all the reviewers for carefully evaluating our manuscript and for their thoughtful suggestions, which have definitely made this a stronger paper. We are delighted that our previous responses and revisions satisfactorily addressed their points.

Reviewer #2 (Remarks to the Author):

The revised manuscript addresses most of my concerns. I have no further comments.